# A Proposed Artificial Intelligence Model for Android-Malware Detection

Fatma Taher [1,*], Omar Al Fandi [1], Mousa Al Kfairy [1], Hussam Al Hamadi [2] and Saed Alrabaee [3]

[1] College of Technological Innovation, Zayed University, Dubai 19282, United Arab Emirates
[2] College of Engineering and IT, University of Dubai, Dubai 14143, United Arab Emirates
[3] College of Information Technology, United Arab Emirates University, Al Ain 15551, United Arab Emirates
* Correspondence: fatma.taher@zu.ac.ae

**Abstract:** There are a variety of reasons why smartphones have grown so pervasive in our daily lives. While their benefits are undeniable, Android users must be vigilant against malicious apps. The goal of this study was to develop a broad framework for detecting Android malware using multiple deep learning classifiers; this framework was given the name DroidMDetection. To provide precise, dynamic, Android malware detection and clustering of different families of malware, the framework makes use of unique methodologies built based on deep learning and natural language processing (NLP) techniques. When compared to other similar works, DroidMDetection (1) uses API calls and intents in addition to the common permissions to accomplish broad malware analysis, (2) uses digests of features in which a deep auto-encoder generates to cluster the detected malware samples into malware family groups, and (3) benefits from both methods of feature extraction and selection. Numerous reference datasets were used to conduct in-depth analyses of the framework. DroidMDetection's detection rate was high, and the created clusters were relatively consistent, no matter the evaluation parameters. DroidMDetection surpasses state-of-the-art solutions MaMaDroid, DroidMalwareDetector, MalDozer, and DroidAPIMiner across all metrics we used to measure their effectiveness.

**Keywords:** malware; deep learning; NLP; android; clustering; static analysis

## 1. Introduction

Android is a popular smartphone operating system with a 70.97% market share [1]. According to the latest statistics [2], there are 2.56 million apps available in official app stores, and there are many more elsewhere. Android leads the smartphone app industry with daily app additions, according to statistics. Android's best feature is its wide choice of feature-rich apps. The popularity of the platform has increased as a result of Google's Play Store adopting an open-source policy for app distribution and providing extensive latitude for app vetting at the time of release.

This popularity and accessibility of app distribution have attracted cybercriminals worldwide. Reports reveal that many mobile malwares target Androids. In the second quarter of 2021, 1.45 million new Android malware apps were detected [3], indicating that new malware is being developed every few seconds. Malicious attacks include DDoS, fuzzing, probing and port scanning [4]. These attacks can endanger transit, application, or other protocols like internet control message protocol, user datagram protocol, simple mail transfer protocol, file transfer protocol, etc. Intrusion detection systems can identify such attacks [5]. Numerous researchers and businesspeople have been thinking about how to prevent malicious apps from penetrating the mobile ecosystem.

Deep learning and machine learning produce intrusion detection systems. Machine learning technology cannot handle the flow of data. Similarly, deep learning algorithms lack optimization, resulting in great generalization mistakes. Fixed Android botnet datasets allow for high-detection-rate detectors [6], but detailed traffic data limits precise prediction.

This has led to the creation of Android-malware classification approaches, which provide the number of neurons and layers during detection [7].

The procedures used to detect malicious Android Application Packages (APKs) often involve both static and dynamic analysis. The dynamic part of the analysis compares the behavior of the application at various points throughout its execution to predetermined test cases. Static analysis, however, involves looking for security flaws in the meta and auxiliary information of the byte code, source code, or application binaries outside of the runtime environment [8]. Due to its high computing cost, the dynamic methodology is not often used for detection due to its reputation for inaccuracy. In addition, unlike the static method, the analysis is conducted after the APKs have already been run. This is why, in terms of generating an initial perspective on the APKs based on their projected behaviors, static analysis is often thought to be faster and more informative.

Before an app is even executed, it can be subjected to a wide variety of methodologies and methods known together as "static analysis", all of which aim to identify its expected behavior under load. For obvious reasons, in a security setting, you would want to identify and remove any apps that have been repackaged or are known to be dangerous before they are installed and run. The authority approximation of an app's likely runtime actions is what static analysis uses to determine whether or not an app is malicious. In most cases, these approximations are the result of techniques such as API calls or code analysis, permissions, app components, intents, native code, file property, etc. [9].

The permission-based security mechanism is used by Android to protect user data and prevent unauthorized apps from accessing private information. One of the most essential security evaluation methods on the Android platform is the permissions given to apps. As a result, without being granted express permission, it is next to impossible to conduct planned action, making permission scanning a crucial stage for malware identification. Before an Android app can do anything useful for its customers, it will ask for access to a variety of systems and data. When combined, several permissions can indicate potentially malicious actions. When an app requests network authorization in addition to SMS access permission, for instance, it may collect users' SMS information and then broadcast it over the Internet. This means that permissions are one of the most popular and useful permanent features in Android.

The survey studies reveal that most Android malware detection researchers do not use feature selection in a comprehensive framework like the proposed one [10,11]. There are numerous benefits to feature selection [12]. The most significant benefit is that feature selection facilitates dimension reduction, which in turn shortens the training phase of classification algorithms. To further improve analytical precision and simplify the model, minimizing the time spent identifying the optimal attributes is crucial [13]. Because of the limitation in the hardware of mobile devices and the need for real-time malware detection systems, feature selection is inevitable. These factors make feature selection-based malware detection systems imperative.

The current research set out to determine how to best go about extracting static characteristics from unknown apps and makes use of a feature selection approach to reduce the most important features. These features reveal if a downloaded app is benign or malicious. Using these characteristics, the efficiency of a deep learning model called the deep learning convolutional neural networks long/short-term memory (CNN-LSTM) technique is evaluated. Malicious apps are then categorized into families through the use of family clustering.

In this paper, we offer DroidMDetection, an effective method for detecting Android malware and grouping it into families by utilizing NLP and deep learning using static analysis features. This study presents the most effective algorithms for keeping an eye on Android apps for signs of hacking. The following are the contributions of the proposed framework:

- To improve the efficacy of the proposed neural networks, we present DroidMDetection, a framework for efficient and accurate malware detection and clustering that makes use

of natural language processing, code static analysis, and machine learning techniques like dropout and feature selection.

- Exploration of various deep learning and machine learning methods for use in developing Android's intrusion detection system.
- Several industry-standard Android datasets were used to test and evaluate the proposed method.
- The tested algorithms are compared to various state-of-the-art models.
- To determine the efficacy and efficiency of DroidMDetection, we conduct an exhaustive evaluation. We analyze DroidMDetection on an obfuscated dataset produced with the PRAGuard [14] dataset and the DroidChameleon [15] obfuscation tool to show the framework's resistance to popular obfuscation methods. We conduct an empirical investigation comparing DroidMDetection to state-of-the-art systems like MaMaDroid [16], DroidMalwareDetector [17] MalDozer [7], and DroidAPIMiner [18], and find that DroidMDetection performs better.

Here is how the rest of the paper is structured. The related work on Android malware detection is introduced in Section 2. The outline for this study is presented in Section 3. In Section 4, the experimental results and performance evaluation are assessed with other related work. Everything is summed up in Section 5.

## 2. Related Work

Because of Android's widespread use and the prevalence of malware, there is a wealth of published studies on the subject. We have selected recently published publications that are of relevance, and we review them here.

### 2.1. Malware Detection

DeepFlow, introduced by Dali Zhu et al. [19], is a malware detection technique built on data streams within malware apps, which may be fundamentally different from those within benign apps but may be similar to other malignant apps to a certain degree. To identify whether or not a new program is malicious, DeepFlow employs a deep learning model that takes into account these differences and similarities.

R. Nix et al. [20] concentrated on program analysis that tracks an app's use of the Android API. API calls are the means by which an app exchanges data with the Android operating system. Such communication is fundamental to an app's ability to perform its functions, and as a result, can reveal critical insights about its behaviors and procedures.

An identification strategy based on Convolutional Neural Network (CNN) was devised and implemented in the system DeepClassifyDroid by Yi Zhang et al. [21]. DeepClassifyDroid's architecture is made up of three parts: a feature extraction component, an embedding in vector space, and a deep learning model that employs convolutional neural networks to classify malware.

To improve categorization efficiency, Alazab et al. [22] created a framework that took advantage of API calls alongside permission requests. To increase the possibility of finding Android malware applications, three distinct grouping algorithms were presented to select the most important API calls. To gauge the efficacy and precision of the suggested strategy when dealing with large datasets, a thorough evaluation of several private and public classes, packages, and methods was carried out.

The DeepRefiner malware identification system, created by K. Xu et al. [23], uses deep neural networks with a wide variety of hidden layers. Before applying any detection rules, DeepRefiner reads XML values from XML files and bytecode from the decompiled classes.dex file. When feeding data into a deep neural network, DeepRefiner represents apps with vectors. Neural systems' hidden layers use the non-linear transformation of input vectors to construct identifying features.

The static analysis just displays the code without actually running it. W. Li et al. [24] developed a deep-belief network-based malware identification system. Threatening API

function calls and permissions were proposed as two features of Android apps that may be used to classify malware.

## 2.2. Android Malware Detection Based on Deep Learning and Machine Learning

According to a recent survey, the number of people who own smartphones is expected to rise substantially from 2019's projected 5.643 billion to 2021's projected 6.378 billion [25]. In addition to making calls and conducting business, people use these devices to share information with one another and communicate socially. Because of the potentially sensitive nature of the data generated by these processes, they must be secured from any unauthorized access [26]. Most assaults on Android devices involve downloading and installing malicious third-party apps, which poses a new problem for security researchers. Several deep learning and machine learning algorithms have been documented in the literature [27] for detecting Android malware. In particular, ref. [28] introduced a back-propagation neural network (BPNN) and convolutional neural network (CNN) hybrid feature-based malware variant identification. However, no statistically engineered aspects have been accounted for in this approach. Similar study by [29] revealed a malware detection accuracy of 93.92% using random forest (RF) and latent semantic indexing (LSI) on the CICInvesAndMal2019 dataset. To solve these problems, the authors of [30] used a non-negative matrix factorization methodology for malware detection, together with feature engineering methods.

To detect malicious apps on Android devices, Zhu et al. [19] presented a stacking integration framework called SEDMDroid. To improve detection accuracy, principal component analysis was applied to each feature subset, with all retained principal components being used to train each multilayer perception model (MLP). After that, we fused the knowledge learned by each member of the ensemble using a support vector machine (SVM) as a classifier.

A unique feature-weighted-based Android malware detection approach, JOWMDroid, was proposed by Cai et al. [31], which coupled the classifier parameters and optimization of weight mapping functions. Following the extraction of eight classes of features from the Android app package, the information gathered was utilized to narrow down the pool of candidates to a manageable set of features optimal for malware detection. Next, they used three different machine learning models to determine an initial weight for each characteristic we'd chosen, and then we used that initial weight as input to one of five different weight mapping functions we developed. Lastly, the differential evolution technique was used to optimize both the weight mapping function and the classifier's parameters simultaneously.

An early malware detection approach based on ensemble behavior was proposed by Aboaoja et al. [32]. The collected evasive behaviors, feature selection and extraction based on correlation, and model development are the three primary stages of the constructed framework. Applying ensemble learning techniques, the framework accurately identified complicated malware activities and made decisions as a result of a majority voting procedure.

To obtain grayscale images, Zhang et al. [33] took things a step further by combining the data portion of AndroidManifest.xml files with DEX files. For Android malware detection, these images are sent to a temporal convolutional network (TCN).

Frenklach et al. [34] also developed a method for the static analysis of Android applications that utilized a similarity graph of the app. With both unbalanced and balanced settings in the Drebin dataset, the brand new VTAz dataset from 2020, and the VirusTotal dataset of over 190,000 programs, the suggested method was shown to be effective, with an area under the curve (AUC) score of 0.987 and accuracy of 0.975 in balanced conditions. The offered approaches had analysis and classification times ranging from 0.08 s/app to 0.153 s/app.

A second Android malware detection solution that focuses on permissions was presented in 2021 by Mahindru and Sangal [35]. Support vector machine with least squares is

one of ten feature selection methods used by the proposed system. In tests, the proposed system demonstrated a 98.8 percent detection accuracy within 12 s. When compared to our proposed approach, the detection time indicates extremely slow processing.

Hei et al. [36] introduced Hawk, a new malware detection system for evolving Android applications, in 2021. Hawk utilized Android's semantic meta-structures for establishing implicit higher-order links as it characterized Android's entities and behavioral relationships as a heterogeneous information network. Over 7 years, the trials analyzed more than 80,860 harmful and 100,375 benign apps. Hawk's out-of-sample application detection averaged 3.5 ms, and its accuracy versus baselines was high.

Mahindru et al. [37] introduced MLDroid, a web-based Android malware detection tool that can identify malicious apps by their access to system resources and their use of APIs. They trained MLDroid using several different machine learning algorithms, including unsupervised, supervised, hybrid, and semi supervised methods, which resulted in improved detection rates.

There have been studies that have used different DL methods to improve Android malware detection systems' effectiveness. Capsule layers were utilized in place of pooling layers in CNNs, as demonstrated by Zhang et al. [38]'s proposed network architecture. Chimera Schranko de Oliveira and Sassi [39] employed multimodal DL, which included a DNN, TN and CNN to learn features from images transformed from the DEX files, static data like permissions, Android intents and dynamic data like sequences of system calls [40].

When it comes to identifying malicious software on Android devices, Yadav et al. [41] provide a comprehensive evaluation of 26 pre-trained CNN models. In total, eight alternative models—VGG19, VGG16, InceptionV3, ResNet50, DenseNet121, MobileNetV2, EfficientNetB4 and DenseNet169—were used for the analysis. We also compared RF and SVM classifiers to these models. Binary classification accuracy was 97% using the proposed technique.

The CNN model may be useful for detecting malware in Android apps, and Martin et al. [42] introduced a novel approach to locating these spots inside the opcode sequence of an app. On the standard-setting Drebin [43] data set, CNN was shown to prioritize areas that were also highlighted by LIME, the gold standard for highlighting feature importance. In addition, the trial outcomes were to one's liking, with a precision = 0.98, accuracy = 0.98, F1-Score = 0.97 and recall = 0.98.

CNN and LSTM were combined by Hosseini et al. [44] to form a hybrid model.lib.so, Classes.dex were extracted from the provided apk archives, and then call graphs were constructed for both of them. The results of the studies showed that the combination of CNN and LSTM achieved a higher rate of accuracy (98.80%) than any of the other machine learning models tested. Methods based on call graphs are restricted to obfuscation strategies like junk codes and unreachable calls.

Using a behavioral model of malware as a series of abstract API calls, MaMaDroid [45] can identify malicious apps. It relies on a mechanism for static analysis to gather API calls performed by an app and then construct a model using Markov chains based on the sequences acquired from the call graph. This makes the model more robust to API changes and keeps the feature set at a manageable size. An F-measure of 99% was achieved when testing MaMaDroid on a dataset consisting of 8500 benign apps and 35,500 malwares collected over six years.

Imtiaz et al. [46] presented DeepMAMADROID, a deep ANN-based method, for both malware identification and detection on Android. Analysis of both static and moving layers is used in DeepMAMADROID. The dynamic base layer will label an application as malicious if the static base layer detects any malicious behavior. With DeepMAMADROID, we were able to classify Android malware with a 93.4% accuracy rate.

By constructing adjacency matrices, Admat [47] can treat each Android app like an "image" for the sake of malware identification and categorization. After the matrices were built, they were fed into the suggested CNN model. Experiments showed that Admat has an accuracy of 98.26% in detecting Android malware.

While previous research has explored the use of natural language processing (NLP) techniques for feature selection in Android malware detection, these approaches often face limitations in capturing the comprehensive nature of malware behavior. In this study, we introduce a novel framework called DroidMDetection that combines multiple deep-learning classifiers with the integration of API calls and intents. This unique combination allows for a more comprehensive analysis of Android malware, going beyond the traditional focus on NLP-based feature selection. By incorporating API calls and intents, our framework captures fine-grained behavioral patterns and interactions, providing a more precise and dynamic approach to malware detection. Moreover, DroidMDetection utilizes deep auto-encoder-generated digests of features to cluster detected malware samples into distinct family groups. This clustering approach further enhances our framework's ability to identify and categorize different families of malware, facilitating targeted analysis and response. By leveraging both feature extraction and selection methods, including deep learning techniques, API calls, intents, and feature clustering, DroidMDetection presents a holistic and advanced framework for Android malware detection. This comprehensive approach sets it apart from previous works solely relying on NLP-based techniques, enabling a more thorough examination of malware characteristics and improving overall detection accuracy.

We acknowledge the importance of the ongoing discussions and research on the challenges posed by the fast evolution of malware ecosystems. Our study contributes to this discussion by proposing a framework that combines deep learning classifiers and feature selection approaches to address the problem of Android malware detection. We believe that this work opens up avenues for further research and advancements in the field, aiming to mitigate the challenges associated with the dynamic nature of malware [48,49].

## 3. The Proposed DroidMDetection Approach

This section describes DroidMDetection, a novel Android malware detection framework written in Python. The framework covers the entire malware detection process. The components of the proposed framework are displayed in Figure 1. The proposed framework consists of six main steps. It starts with feature extraction applied to extract static features using static analysis. Then, preprocessing to normalize the values of datasets to be in the same scale. Then, feature selection is applied to reduce the dimensionality of the datasets to reduce the training and testing execution time. Feature vectorization puts the textual features into vector form. Malware detection is then applied to classify the apps into both malicious and benign using different classification models. Finally, malicious apps are then clustered into similar groups of malwares. The next subsections provide a discussion for all the proposed model stages in details.

### 3.1. Feature Extraction

Android apps are packaged and installed in the Android Package Kit (APK) file format. It has everything a user needs to get an app installed on their device. This document contains a wide variety of information, such as API calls, application source codes, images and permissions. You can think of APK files as compressed files. Therefore, APK files need to be opened to extract the required data from the app's files. In our experiments, AAPT2 is used to access application-specific files [50]. As an attribute, applications' access to the AndroidManifest.xml file reveals which permissions they require. AAPT2 combines static analysis capabilities with the ability to generate Dalvik assembly code. Dalvik assembly is generated by the Dalvik compiler when it translates Android app bytecode into executable instructions for the Dalvik VM. AAPT2 enables developers to analyze and extract information from Android app resources, such as layouts, strings, and assets. To avoid abusing access, the feature vector is built taking into account just the permissions that come as standard with Android apps [51]. Algorithm 1 describes how to process APK files and generate the feature vector. From these data, we were able to extract the following features:

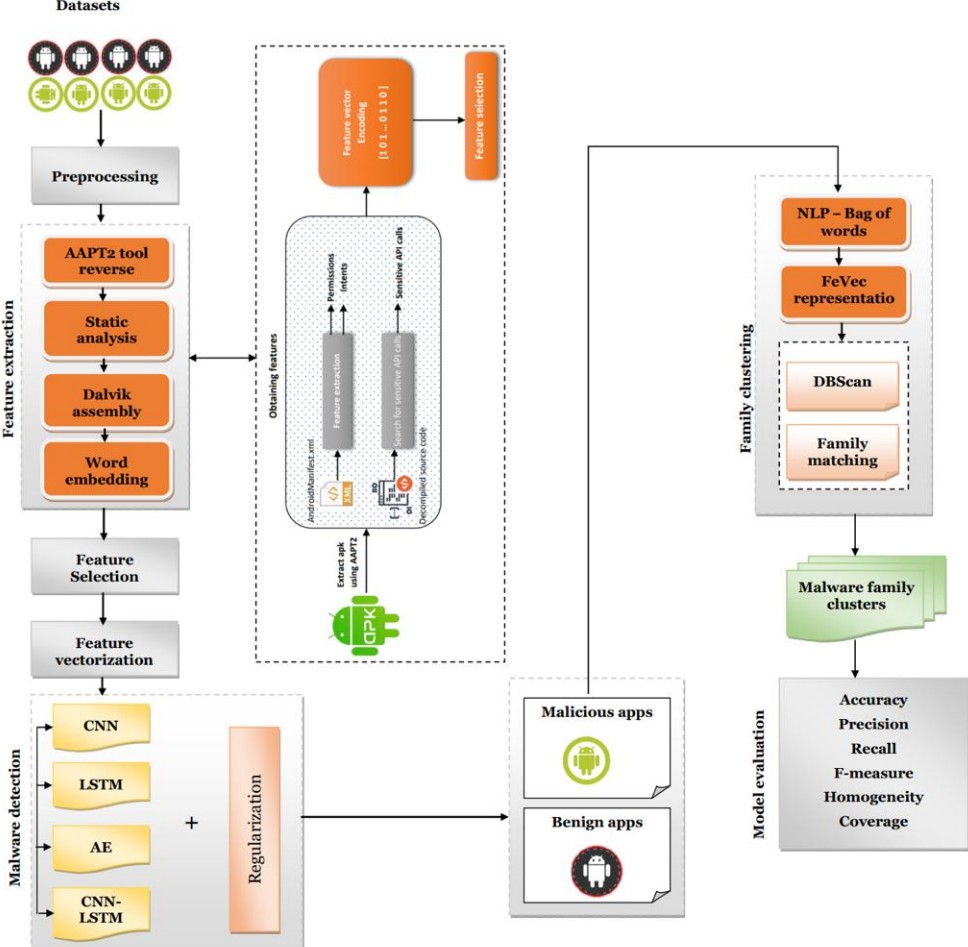

**Figure 1.** The Proposed DroidMDetection framework.

- Permissions: A large range of permissions, such as READ_CONTACTS, CAMERA, and CALL_PHONE, is provided as the Android Application Programming Interface (API) to determine the powers of Android apps. This means that an app's functionality is constrained by the rights it has been granted, as specified in the AndroidManifest.xml file. Due to the availability of the open-source reverse engineering tool for Android apps known as apktool [52], apk archives could be decompiled and analyzed. The AndroidManifest.xml file for each app was then parsed for its set of permissions.
- Intents: They are what characterize communications between apps in the Android ecosystem [53]. A malware's goals can be encoded using intents, which are semantically rich features [54]. Intents, like permissions, are declared and retrieved from the AndroidManifest.xml file.
- The source code must be analyzed to do extra checks alongside these static features concerning maliciousness. By utilizing apktool, not only the AndroidManifest.xml file, but also the decompiled source code in smali format and the disassembled form of the DEX format used by Android's Java VM implementation are extracted during the reverse engineering process [55]. Table 1 details the APIs that DroidMDetection identified as potentially malicious. To find the sensitive API calls, the decompiled code was examined recursively.

**Table 1.** API calls that DroidMDetection deemed sensitive.

| | |
|---|---|
| getSubscriberId | getBondedDevices |
| getDeviceId | startDiscovery |
| getSimSerialNumber | abortBroadcast |
| getLine1Number | setWifiEnabled |
| getAllCellinfo | setPreviewDisplay |
| getCallState | MediaRecorder |
| getAccounts | createFromPdu |
| getNetworkInfo | sendMultipartTextMessage |
| getExtraInfo | sendTextMessage |
| requestLocationUpdates | obtainMessage |
| getLastKnownLocation | sendDataMessage |
| getSimOperator | killProcess |
| getNetworkOperator | myPid |
| getNeighboringCellInfo | exec |
| getCellLocation | createSubprocess |
| DexClassLoader | |

In Figure 2, we can see the entire feature extraction process of DroidMDetection.

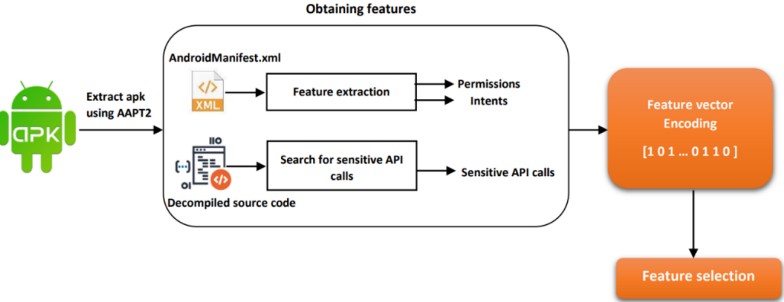

**Figure 2.** DroidMDetection's feature extraction workflow.

Applying Algorithm 1, we can determine which features are used by every application in the dataset, and then build a feature vector based on that data. In Figure 3, we can see a portion of the feature vector. Assigning a value of 1 if the app requests a necessary authorization and a value of 0 otherwise transforms the feature vector into a mathematical structure. Some characteristics in the obtained feature vector are found to be of no use in any of the applications. When these are taken out of the feature vector, only 102 features remain for analysis. The application-agnostic processing dataset, of which Figure 3 displays a subset, is the result.

```
1,1,0,0,0,1,0,1,1,1,1,0,0,1,1,0,1,1,0,0,
0,0,0,0,0,0,0,0,1,1,1,1,0,0,0,0,1,1,0,0,
1,1,0,0,1,0,1,0,0,0,1,0,1,0,0,0,0,0,0,1
1,0,0,0,0,1,0,0,0,0,0,1,1,0,0,0,0,0,0,0,
0,1,0,0,1,1,1,0,0,0,0,0,0,0,0,0,0,0,1,
0,0, malware
0,0,1,1,1,0,1,1,0,0,0,0,0,0,0,1,1,0,1,
1,0,1,0,1,0,1,0,0,0,0,0,0,0,1,0,0,0,1,
1,1,0,0,0,0,0,0,0,0,0,0,0,1,0,0,0,0,0,
0,0,1,0,0,1,0,0,0,0,0,1,0,0,0,1,0,0,1,
1,0,0,1,0,0,0,0,0,0,0,1,1,0,0,0,0,1,0,1,
1,0, benign
```

**Figure 3.** A portion of the dataset after processing.

---

**Algorithm 1:** Extracting the feature vector

---

**Input:** All apps and standard features.
**Output:** *feature_Vec*

  1.  **def** FeatureVector(*apps, features*):
  2.      *feature_Vec*[][] ← ϕ
  3.     $M_1$ ← *length*(app)
  4.     **for** $i$ ← 1 **to** $M_1$ **do**
  5.       invoke AAPTT2
  6.       extract app[$i$] using AAPT2
  7.       obtain app[$i$]. AndroidManifest.xml
  8.       $M_2$ ← length(standardfeature)
  9.       **for** $j$ ← 1 **to** $M_2$ **do**
 10.         **if** *standardfeature[j] is member of app[i].*
    *AndroidManifest.xml*
 11.         **then**
 12.           *feature_Vec[i][j]* ← 1
 13.         **else**
 14.           *feature_Vec[i][j]* ← 0
 15.         **end if**
 16.     **end for**
 17.   **end for**
 18.   **return** *feature_Vec*
 19. **End def**

---

### 3.2. Preprocessing

Due to the varied nature of the Android datasets, pretreatment is essential for effective data management.

Min–Max Normalization Method

The values in a dataset can be shifted and rescaled with the use of a technique called normalization. The data were normalized between 0 and 1 using the min–max technique. The following equation was used to normalize the full dataset's overlap using the normalization method [56]:

$$\widehat{a} = \frac{a - x_{min}}{max(A) - min(A)}(\text{newmax}(A) - \text{newmin}(A)) + \text{newmin}(A) \tag{1}$$

where *max(A)* and *min(A)* are the maximum and minimum data, respectively, "newmax" (*A*) and "newmin" (*A*) are the new values of the maximum and minimum used for the data scaling, and $\widehat{a}$ is the normalized data.

### 3.3. Feature Selection

Our paper contributes significantly to the field by demonstrating how to choose a smaller set of attributes without sacrificing the efficacy of a malware detection system [57]. Singular value decomposition, principal component analysis, and linear discriminant analysis are just some examples of dimensionality reduction algorithms, and they give rise to two main worries. The first is that only the features that are used to train the classifier undergo a decrease in quantity. Data collection is unaffected; therefore, there is no loss of productivity. The second problem is that in practical deployments, all of these methods need resource-intensive preprocessing of the gathered data to prepare it to be fed into the classifier. Because of this, we choose to use linear regression. Because of this technique, not only was the dimensionality of the data coming into the system decreased, but the number of features needed for prediction in production environments was too. This allowed for faster data collection, training, and testing, as well as less cumbersome deployment in the world. Modelling the association between two or more variables is the goal of linear regression, a statistical technique [58]. If only one independent variable is used in the

created model to assess the dependent variable, we refer to this as simple regression; if numerous independent variables are included, we refer to this as multiple regression. Application features serve as the independent variables in this analysis, with application types serving as the dependent variable. Regression models require a numeric dependent variable.

Thus, in Figure 3, labels for malicious apps are set to 0, while labels for benign apps are set to 1. Regression analysis is used to determine the strength of association between a dependent variable and a set of independent variables. In a linear model, the connection between the dependent variable and n independent variables looks like Equation (2):

$$Y = b_0 + b_1 X_1 + b_2 X_2 + \cdots + b_n X_n + e \tag{2}$$

where the dependent variable is denoted by $Y$, and the coefficient used in the model is referred to by $b_1, b_2, \ldots, b_n$. The application features are presented by $X_1, X_2, \ldots, X_n$. The point where the y-axis is intersected is $b_0$, and the error is shown as parameter $e$. The least squares approach is used to determine these coefficients. The prediction error in Equation (2) is attempted to be minimized by employing the least squares method.

$$SSE = \sum_{i=1}^{n} (y_i - \breve{y}_i)^2 \tag{3}$$

where the total number of data is represented by $n$, the predicted/estimated value of the model is denoted by $\breve{y}_i$, and $y_i$ is the actual data.

The Sum of Squares for Error (SSE) is the total squared error in a set of predictions. Once the coefficients are differentiated, the SSE value is adjusted to be as close to zero as possible in linear regression. The resulting model is the multiple linear regression model depicted by Equation (2). By utilizing the least-squares method, as shown in Equation (4), a multiple linear regression model can be obtained with three coefficients and two independent variables.

$$\begin{bmatrix} n & \sum_{i=1}^{n} x_{1,i} & \sum_{i=1}^{n} x_{2,i} \\ \sum_{i=1}^{n} x_{1,i} & \sum_{i=1}^{n} x_{1,i}^2 & \sum_{i=1}^{n} x_{1,i} \cdot x_{2,i} \\ \sum_{i=1}^{n} x_{2,i} & \sum_{i=1}^{n} x_{1,i} \cdot x_{2,i} & \sum_{i=1}^{n} x_{2,i}^2 \end{bmatrix} \begin{bmatrix} a \\ b \\ c \end{bmatrix} = \begin{bmatrix} \sum_{i=1}^{n} y_i \\ \sum_{i=1}^{n} x_{1,i} \cdot y_i \\ \sum_{i=1}^{n} x_{2,i} \cdot y_i \end{bmatrix} \tag{4}$$

where $y$ represents the dependent variable. Variables $x_1$, $x_2$ denote the independent variables, and the coefficients are represented by $a$, $b$ and $c$.

By computing the coefficients for each independent variable, we can see how much of an effect they have on predicting the dependent variable. Multiple linear regression models built from the collected data show that the coefficients range from $-1$ to 1. Taking a look at Figure 3, we can tell that the processed dataset is primarily a sparse matrix made up of 0 s. Some permission or feature coefficients will be zero or very close to zero under these conditions. In this way, the permissions with coefficients close to 0 and 0 are removed from consideration during the feature selection process using Algorithm 2.

*3.4. Feature Vectorization*

The features are then encoded in a one-hot feature vector, as described in Section 3.2. According to studies in the Natural Language Processing field, one-hot encoding does not include any corpus information, and the distance between any two words is the same. Word2vec [59] builds a word vector based on context, with highly relevant words being closer together. In other words, word2vec is more evocative and better able to communicate the underlying qualities of data. Word2vec uses either the Continuous Bag-of-Words (CBOW) model or the Skip-Gram model to generate a dispersed representation of words. The model in the continuous bag-of-words architecture makes predictions about the next word based on a window of previously predicted words. The current word is used to make predictions about a window of adjacent words in the context learning model in skip-gram architecture.

---

**Algorithm 2:** Obtaining deleted features

---

**Input:** Linear model of regress.
**Output:** Del.Feature_Set

1. **def** FeatureSelection(*RegressModel*):
2.     Convert the linear regression model to pairs of coefficients and independent variables.
3.     Store the pairs obtained in a hash data structure called *SelectModel*{}
4.     $M_1 \leftarrow length$(SelectModel)
5.     **for** $i \leftarrow 1$ **to** $M_1$ **do**
6.         **if** *SelectModel[SelectModel.keys[i]]* < 0.1 **then**
7.           **if** *SelectModel[SelectModel.keys[i]]* > −0.1 **then**
8.             *Del.Feature_Set* ← *SelectModel.keys[i]*
9.           **end if**
10.         **end if**
11.     **end for**
12.     **return** *Del_Feeature_Set*
13. **End def**

---

To better express features relevant to Android malware classification, we attempt to apply word2vec, which uses word embeddings. In this work, we conceptualize features derived from Android application packages as words. Specifically, a k-dimensional vector is used to represent each feature. When it comes to training, we rely on the CBOW model. Assuming N samples, X feature dimensions and K word vector dimensions, the final trained matrix has N × K × X) dimensions. Every point in the k-dimensional space can be represented by a vector, and the elements of each vector are learned through repeated training and feature-weighting iterations. Throughout all of our models, we set K = 100. The details of the vectorization procedure are shown in Algorithm 3.

First, we use the method described in the previous section to choose features in lines 3 and 4. On line 5, we have separated the feature document into four sentences based on the feature type. Thirdly, the word vector is obtained via training a word2vec model in lines 6–7. Lastly, in lines 8–14, we substitute 0 for any attributes that are not present in the sample.

---

**Algorithm 3:** Vectorization of features

---

**Input:** Sample feature documents **S**
Dimension of word2vec **K**
Feature list **L** of number **N**
**Output:** M × (K × N)- dimension vector as **V**

1. **def** FeatureVectorization(*S, K, L, N*):
2.     **foreach** $s_i \in S$ **do**
3.       sentences = *empty_list*()
4.       $s_i$ = feature_select($s_i$)
5.       Sentences ← extract sentences from $s_i$
6.       model = word2vec_train(sentenses, K)
7.       word_dict ← model.wv.vocab
8.       zero_vec ← K-dimension zero vector
9.       **foreach** $l_i \in L$ **do**
10.         **if** $l_i \in word.dict$ **then**
11.           V.append(word_dict[$l_i$])
12.         **else**
13.           V.append(zero_vec)
14.         **end if**
15.       **end foreach**
16.     **end foreach**
17.     **return** *V*
18. **End def**

---

### 3.5. Malware Detection

Both LSTM and CNN are algorithms from deep learning, and when combined, they form a fusion model known as CNN-LSTM. CNNs use hidden neurons whose weights and biases can be adjusted through training. Unlike other structures, it is often used to analyze data in a grid format [60]. Since the data flow in only one direction from the input to the output, this type of network is also known as a feed-forward network. Convolutional, pooling, and fully connected layers make up the bulk of a CNN's architecture. In deep learning, the convolutional and pooling layers are used for feature extraction and dimensionality reduction. Folded and connected to the preceding layer's output, the fully connected layer is the final stage of the stack. Figure 4 depicts the primary architecture of the CNN model used to identify malicious Android apps.

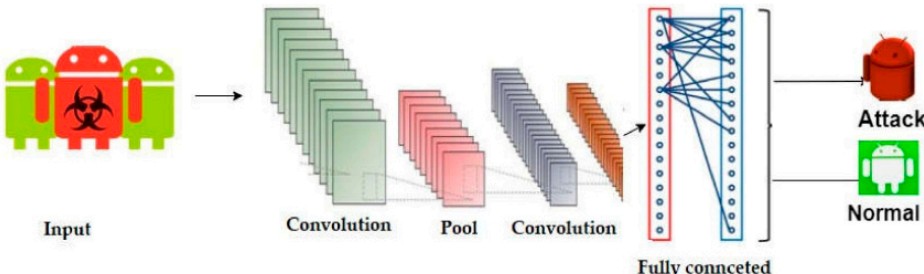

**Figure 4.** CNN model's structural elements.

LSTM is an approach for learning long-term data dependency, and it was first presented by Hochreiter et al. [61]. One kind of recurrent neural network (RNN) is long short-term memory (LSTM). In contrast to RNN methods, the LSTM architecture includes memory cells. Candidate, input, output and forget make up the four parts of a memory cell. The input features are classified as "forgettable" or "keepable" by the forget gate. The LSTM structure's input gate refreshes the memory cells, while the LSTM's output gate maintains in constant control over the hidden state. In addition, LSTM handles difficulties with the RNN learning's disappearing gradient and explosion gradient by employing a gate mechanism and embedded memory block [62]. Figure 5 depicts the basic layout of the LSTM model.

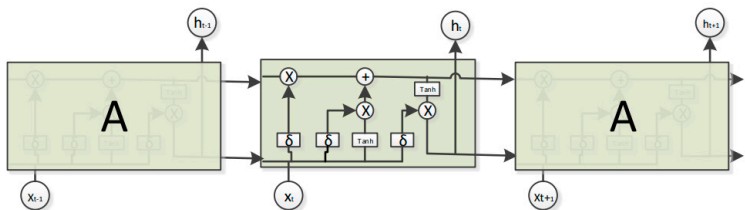

**Figure 5.** The LSTM technique's structure.

The LSTM model's settings can be found in Table 2. Researchers found that these settings were crucial for achieving optimal performance in malware detection for Android. When picking out relevant features from the filter layer, the maximum pool size was 4, and the size of the kernel of convolution was 4. To avoid overfitting, we set the dropout value to 0.50, and we show how the function of the RSMprop optimizer may be used to fine-tune the model. A batch size of 150 is employed for the error gradient. The following equations characterize LSTM-related gates:

$$f_t = \sigma\left(W_f \cdot X_t + W_f \cdot h_{t-1} + b_f\right) \tag{5}$$

$$i_t = \sigma(W_i \cdot X_t + W_i \cdot h_{t-1} + b_i) \tag{6}$$

$$S_t = tanh(W_c \cdot X_t + W_c \cdot h_{t-1} + b_c) \tag{7}$$

$$C_t = (i_t \cdot S_t + f_t \cdot S_{t-1}) \tag{8}$$

$$o_t = \sigma(W_o + X_t + W_o \cdot h_{t-1} + V_o \cdot C_t + b_o) \tag{9}$$

$$h_t = o_t + tanh(C_t) \tag{10}$$

where the weight matrices are denoted by $W_f, W_i, W_o, W_c$, and $V_O$. The input features' vector is referred by $X_t$.

**Table 2.** Setting the parameters for the LSTM model.

| Items | Configuration Value |
|---|---|
| Max pooling size | 4 |
| Kernel size | 4 |
| Fully connected layer | 32 |
| Epochs | 20 |
| Dropout | 0.50 |
| Activation function | Relu |
| Optimizer | RSMprop |
| Batch size | 20 |

The obtained values for the output, forget and input gates are, at any time $t$, as follows: $o_t, f_t, i_t$. The short memory vector is represented by $h_{t-1}$. The point at which the memory cell's declared value at time $t$ is $h_t$. The bias vectors are represented by $bi$, $bc$, $bf$, and $b0$. $tanh$ and $\sigma$ are the activation functions. At time $t$, the memory cell's candidate value is represented by $St$, and the memory cell's state is represented by $Ct$.

As can be seen in Figure 6, a CNN-LSTM model was developed. It was taught on the training data, and then its hyper parameters were fine-tuned with the help of the validation data and the Adam optimizer. Next, The test dataset was run through several models, including the CNN, AE, LSTM and CNN-LSTM model, which mapped each testing tuple's features to its true class (benign or malicious) [63]. The experiments in Section 4 proved the superiority of CNN-LSTM over other classifiers in android malware detection. Therefore, the rest of the evaluations are conducted using CNN-LSTM. Two one-dimensional convolution layers with a kernel size of 4 and 32 filters make up the CNN-LSTM model that is used for training and optimization, with two fully connected layers composed of an output layer with the SoftMax activation function and 256 hidden neurons. The overfitting issue was fixed by using a combination of dropout and global max-pooling layers. Using the global max-pooling layer, which captures the maximum value, and the dropout layer, which turns off a portion of the CNN-LSTM network's neurons, overfitting of the learned features can be prevented. Adam optimizes by making changes to the weights to reduce the loss function. The CNN-LSTM model's settings are displayed in Table 3.

**Table 3.** Setting up the CNN-LSTM model parameters.

| Items | Configuration Values |
|---|---|
| Max pooling size | 4 |
| Fully connected layer | 32 |
| Kernel size | 4 |
| Drop out | 0.50 |
| Optimizer | RSMprop |
| Activation function | Relu |
| Epochs | 20 |
| Batch size | 150 |

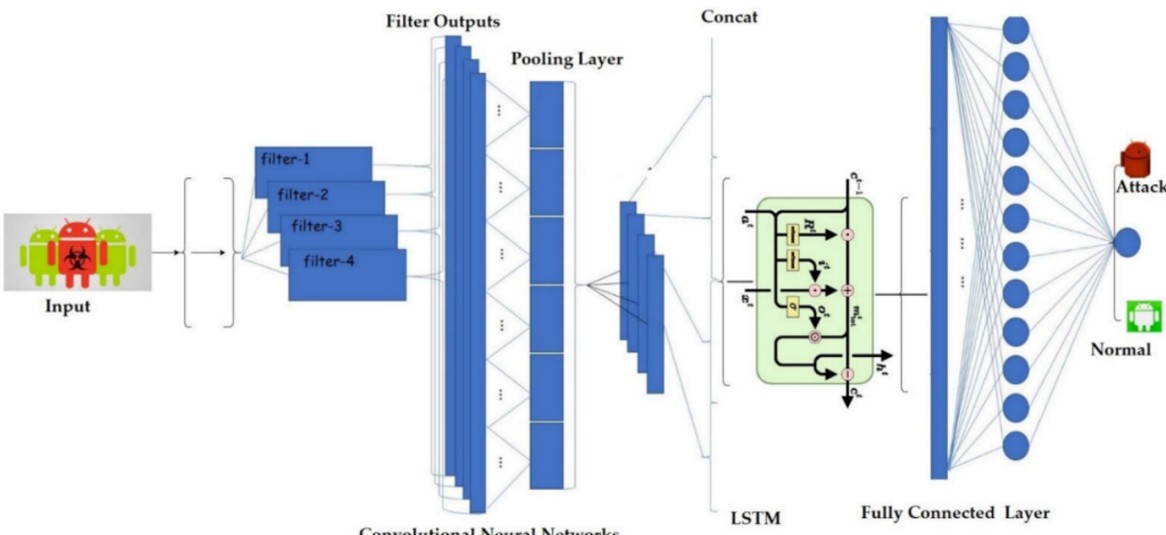

**Figure 6.** Structure of the CNN-LSTM model.

*3.6. Malware Clustering*

Malware family clustering, in contrast to classification, necessitates an additional representation that compresses an instruction sequence into a single feature vector rather than a sequence of embedding's for a specific malware sample. With the use of the feature hashing technique [64] and a bag of words (N-grams) NLP model, we generate feature vectors for each malware sample found by analyzing the code's instruction sequences. Each result is a vector referred to as a FeVec vector, and there is one for every piece of malware that was detected. Using deep neural auto-encoders [65] on the FeVec vectors, we generate an even more compact digest or embedding for each malware sample. The malicious apps that have been detected by DroidMDetection are grouped into families based on the similarities in their digests. Clustering in DroidMDetection is performed using the DBScan [66] method.

This section goes over the system of family clustering in detail. The goal of the clustering phase is to categorize the malicious apps discovered in the previous phase into subsets that share a high degree of similarity and are therefore likely to be related strains of malware.

3.6.1. Android App Representation for Clustering

We represent the Dalvik assembly as a collection of code snippets, each of which represents a method from a single class. Because Dalvik code DC is structured as a collection of classes, this division makes sense $DC = \{C_1, C_2, \ldots, C_s\}$. Methods are organised into classes, with each class $C_i$ having its own $C = \{R_1, R_2, \ldots, R_k\}$ where the actual instructions reside. An Android app's micro-behavior may involve the execution of a method, while the app's macro-behavior may involve a global execution path. An Android app can have many global execution pathways determined by external events. Malware on Android, however, is typically designed with several others and one essential global execution path to trick anti-malware tools. The malware may generate distinct global execution paths for the payload. However, micro-behavior is still necessary for other macro-behavior. Preprocessing the assembly resulting from DroidMDetection yields many sequences $H = \{S_1, S_2, \ldots, S_h\}$, in which every sequence $S$ is a set of instructions $S = \langle A_1, A_2, \ldots, A_v \rangle$ of a method of a class. In other words, $H$ comprises sequences' instruction $H = \{\langle A_1, A_2, \ldots \rangle_1, \langle A_1, A_2, \ldots \rangle_2, \ldots, \langle A_1, A_2, \ldots \rangle_h\}$.

3.6.2. Clustering Preprocessing

The clustering procedure begins with a set of sequences, denoted $H = \{S_1, S_2, \ldots, S_h\}$, of the malicious apps that were found. To create malicious apps' embedding digests,

we present a new technique based on natural language processing and a deep neural network auto-encoder called FeVec. Next, we use the DBScan clustering algorithm to classify the malware samples into families by clustering their digests. Take note that unlike the classification phase, in which a list of embeddings is sufficient to represent malware samples, our clustering technique requires a single feature vector to represent each sample. Because of this, we describe a novel method called FeVec, which uses automated feature vector representation to identify malware samples. The FeVec method utilizes feature hashing [67] and an NLP bag of words (N-grams) [68–71] to form fixed-size embeddings from concatenated instruction sequences. This step consists of two main steps.

The N-gram has seen widespread use in authorship attribution, automatic text classification and other applications such as text analysis and natural language processing. For a big sequence, n-gram can be used to calculate the sequences of n items that are consecutive. In the proposed framework, N-grams of instructions (of length n) are calculated for a concatenated sequence P. Note that a forward-moving window (of size n) is used to extract the N-grams, and that the counter of the detected sequence is incremented by one after each iteration. Due to our experiments, we find that the optimal value for the window-size hyper-parameter, n, is 4. Our findings show that the effectiveness of feature vector generation is drastically impacted when n is larger than 4. The efficiency of the clustering changes when n < 4. To avoid excessive use of memory and computation caused by the high dimensionality of N-grams, we compute them concurrently with the feature hashing.

To vectorize P, DroidMDetection uses N-grams and Feature Hashing (FH) that accepts the feature vector's target length Q and N-grams of P as inputs. The final result is a feature vector of size Q consisting of components $y_i$. We set Q = |V| in our framework, where V is a vocabulary list. With Formula (11), we see that the Euclidean norm is used to normalize $y_i$. FeVec generates a hashing vector, $hashV$, of fixed size from the signature of a malicious software P. So, the hashing vector $hashV = \{hashV_0, hashV_1, \ldots, hashV_{DM}\}$ is the result of DM malicious apps that were detected before.

$$L2Norm(y) = \parallel y \parallel_2 = \sqrt{y_1^2 + \ldots + y_n^2} \tag{11}$$

The square root of the sum of the squares of the vector values is the euclidean norm. Extensive prior study [59] demonstrates that the hash kernel roughly maintains the vector distance and scales linearly with the sample size.

### 3.6.3. Digests Generation Using Auto-Encoder

We create a deep neural auto-encoder by stacking neural layers and performing encoding and decoding operations. Latent app representations are learned unsupervised by the proposed auto-encoder. Auto-encoders can learn unsupervised when their input data are reconstructed from hash vectors $HAV = \{hashV_0, hashV_1, \ldots, hashV_{DM}\}$ that have not been labelled (Table 3). It is important to note that when training the auto-encoder used in DroidMDetection, we do not need any labelling, as data from publicly available Android applications are sufficient.

The training's goal is to teach the auto-encoder how to consistently produce a digest of an Android app's hashV that maintains the traits that set malicious apps apart from benign ones. Formally, an unlabeled hash vector $HAV = \{hashV_0, hashV_1, \ldots, hashV_{DM}\}$ is sent into the deep neural auto-encoder network as input-referred as $T' \in \mathcal{U}$, where the encoder circuit operates $f_{encoder} : \mathbb{R}^{|V|} \to \mathbb{R}^a$ for, $a = 64$ which is parameterized by $\Theta_{encoder}$ to form the digest $I_{T', \Theta_{encoder}}$

$$I_{T', \Theta_{encoder}} = f_{encoder}(T'; \Theta_{encoder})$$

The decoder circuit $f_{\text{decoder}} : \mathbb{R}^a \to \mathbb{R}^{|V|}$ uses the resulting digest, $\boldsymbol{I}_{\boldsymbol{T}',\Theta_{encoder}}$, to rebuild the FeVec feature vector. The auto-encoder circuit's training loss, given $\boldsymbol{T}'$, is:

$$\widetilde{\boldsymbol{T}}' = f_{decoder}(\boldsymbol{I}; \Theta_{decoder})$$

where the generated reconstruction is denoted by $\widetilde{\boldsymbol{T}}' \in \mathbb{R}^{d \times w}$.

$$\mathcal{L}_{auto\text{-}encoder}\left(\boldsymbol{T}'; \Theta_{encoder}, \Theta_{decoder}\right) = \| \boldsymbol{T}' - f_{decoder}\left(\boldsymbol{I}_{\boldsymbol{T}',\Theta_{encoder}}; \Theta_{decoder}\right) \|^2$$

During training, an unlabeled Android app's FeVec feature vectors have their objective reconstruction function minimized using a gradient-based optimizer.

$$\left(\Theta^*_{encoder}, \Theta^*_{decoder}\right) = arg \min_{\Theta_{encoder}, \Theta_{decoder}} \sum_{i=1}^{M1+M_2} \mathcal{L}_{auto\text{-}encoder}\left(\boldsymbol{T}'_i; \Theta_{encoder}, \Theta_{decoder}\right)$$

Due to its widespread usage, the auto-encode used by DroidMDetection only needs to be trained once before being put to use in any of the experiments. Specifically, DroidMDetection uses a trained encoder, $f_{decoder}$, to generate digests $I = \{d_0, d_1, \ldots, d_{DM}\}$ for the malicious apps detected in the previous step.

3.6.4. Family Clustering

Using a clustering algorithm, DroidMDetection clusters and organizes the malware digests $D = \{d_0, d_1, \ldots, d_{DM}\}$ into families based on their shared characteristics. When it comes to clustering in DroidMDetection, first and foremost, only samples with a high degree of similarity are put into clusters, while the rest are labelled as nonclustered by the clustering algorithm. We may not always find malicious apps from the same family, and we would prefer to have family groups only if the sample malware family includes groups, so this functionality may be more useful in real-world deployments. We use the DBScan clustering technique to implement this function. A second, discretionary phase involves selecting the optimal cluster for the non-cluster samples from among the clusters generated by calculating the euclidean similarity between a specific cluster sample and a specific non-cluster sample. This process is known as "family matching". The evaluation includes both pre- and post-optional step homogeneity and coverage metrics for the clustering. In contrast to K-means and other clustering algorithms, DBScan generates highly reliable clusters. The homogeneity of the produced clusters is the most crucial statistic in DroidMDetection clustering.

## 4. Experimental Results

In this section, the evaluation of the proposed framework is evaluated. Malware detection process and clustering of different families are assessed with other related researches.

### 4.1. Implementation Environment

DroidMDetection is written in Python. To convert DEX bytecode to Dalvik assembly, we utilize the tool dexdump5. Dexdump is an easy-to-use but powerful program for extracting textual disassembly from APK files. Take note that the preprocessing has not been optimized; just one thread script is used to evaluate the app's efficiency. Specifically, we use PyTorch6 to carry out DroidMDetection activities. We use the standard hdbscan7 implementation for clustering. The implementation computer's hardware and software requirements are listed in Table 4. The classifier's preliminary processing, training, and testing were all conducted on this machine. It also served as a repository for a simulated testing environment.

**Table 4.** Hardware and software for the environment of the implementation.

| HW/SW | Settings |
|---|---|
| Clock speed | 2.40 GHz |
| Processor | Ryzen 5 3600, MAMADROID |
| GPU | NVidia RTX 3060Ti |
| RAM | 64 GB |
| Python | 3.9.14 |
| Operating system | Windows 10 |
| Sci-Kit Learn | 0.24.1 |
| VMWare Workstation Pro | 16.0 |

*4.2. Dataset*

As shown in Table 5, our evaluation dataset includes millions of Android apps gathered over the past decade to serve as a sampling space for our studies. Our study is convincing because of the breadth of size, time, and malware families that it covers. Malware from Drebin [43], MalGenome [69], MalDozer [7], and MaMaDroid [46] are used to test DroidMDetection's family clustering and detection capabilities. In addition, we employ benign programs from the AndroZoo [66] repository. Given that many families have very few data, we focus on the 20 most frequent families, shown in Table 6. The malware samples used in the family clustering examination are taken directly from the reference datasets. The various stages of the experiments were as follows:

(1) Training and testing: To acquire preliminary findings using all five classifiers, the dataset was divided randomly into a 25% testing subset and a 75% training subset [7,11,40,46].

(2) Feature selection: We then narrowed down the features we were using by selecting fewer of them. Specifically, linear regression was the technique of choice for the feature selection phase. To generate new results, the classifiers were retrained using the smaller dataset that resulted from the feature selection process.

(3) Malware detection performance: Classification results are obtained and evaluations are reviewed based on performance metrics after the most important features have been selected. The proposed model for android detection uses a large number of classifiers to guarantee its generalizability.

(4) Family clustering: Once malicious apps have been identified, clustering is used to categorize them into like-minded groups.

Table 7 explains the confusion matrix.

*4.3. Evaluation*

Here, we analyze the DroidMDetection framework using a variety of tests and configurations on various datasets. The following activities are assessed: (1) DroidMDetection's detection efficacy on both small and big training datasets; (2) the effects of feature selection; (3) the efficacy of a clustering technique that emphasizes families and number; (4) in terms of runtime efficiency on common hardware, how well does DroidMDetection perform? (5) DroidMDetection's resistance to widespread forms of obfuscation.

**Table 5.** Used datasets and example of family names.

| Name | Number of Families | Number of Samples |
|---|---|---|
| Drebin [35] | 179 | 5.5K |
| MalGenome [59] | 49 | 1.3K |
| MaMaDroid [37] | - | 40K |
| MalDozer [7] | 20 | 21K |
| AndroZoo [60] | - | 9.5M |

**Table 6.** Family names.

| Family | Family |
|--------|--------|
| GoldDream | BaseBridge |
| GinMaster | Adrd |
| Imlog | DroidKungFu |
| Iconosys | DroidDream |
| MobileTx | FakeDoc |
| Kmin | ExploitLinuxLotoor |
| Plankton | FakeRun |
| Opfake | FakeInstaller |
| SMSreg | Geinimi |
| SendPay | Gappusin |

**Table 7.** Confusion matrix.

| | | Actual | |
|---|---|---|---|
| | | **Positive** | **Negative** |
| Predicted | Positive | True Positive (*TP*) | False Positive (*FP*) |
| | Negative | False Negative (*FN*) | True Negative (*TN*) |

### 4.3.1. Performance Metrics

The classification of the dataset yields four potential outcomes. These are false negative (*FN*), true positive (*TP*), false positive (*FP*) and true negative (*TN*). True positivity (*TP*) occurs when a sample is appropriately identified as positive. Mislabeling a sample that should be positive as negative is known as false negative (*FN*). The term "*TN*" is used to describe the situation in which a truly negative sample is appropriately identified as negative. If a sample is falsely identified as positive when it is truly negative, we call that an FP. Table 1's confusion matrix includes all of these scenarios. Precision, accuracy, F1-score and recall are used to display the evaluation results. We evaluate the efficiency of family clustering using the coverage and homogeneity [71] measures. The generated family clusters are evaluated on their degree of purity using the homogeneity metric. Since DroidMDetection clustering only seeks to construct groups with certainty while disregarding fewer certain groups, a perfect homogeneity means that each formed cluster contains samples from only one malware family. The percentage of a clustered dataset that may be trusted is measured by coverage metrics. Precision (*P*) is the proportion of correct predictions or the fraction of malicious applications found in a given set of sample apps; $P = \frac{TP}{TP+FP}$.

A system's recall (*R*) indicates how many malware samples were accurately identified as malicious software; $R = \frac{TP}{TP+FN}$.

The number of all accurate predictions divided by the overall dataset size yields accuracy (ACC); ACC = $\frac{TP+TN}{TP+TN+FP+FN}$.

Calculating just the accuracy, recall and precision values is not enough to evaluate the efficacy of classification systems. To measure how well a classification system performs, we use the F1-Score (F1), which is the harmonic mean of recall and precision.

### 4.3.2. Malware Detection

Here, we detail how well DroidMDetection can spot malware, and how changing hyper-parameters affects that detection.

**(1) Detection Performance**

This study compared the effectiveness of the LSTM, AE, LSTM-CNN and CNN, proposed in this study concerning Android malware detection in the model DroidMdetection proposed in this study when used in conjunction with the feature selection approach proposed in this study. These classifiers were specifically chosen for their ability to capture

different aspects of the malware behavior and provide complementary detection capabilities. The primary objective was to compare the effectiveness of these diverse classifiers within the proposed model. Each classifier brings its unique strengths and characteristics to the DroidMDetection model, allowing for a comprehensive analysis of malware samples.

To ensure the generalizability of the proposed model, the experiments are performed on the aforementioned datasets. Tables 8–11 detail the outcomes of the experiments. The outcomes of the LSTM, AE, LSTM-CNN and CNN models are displayed in Table 8. The CNN-LSTM model beat the CNN, LSTM, and AE models with high accuracy on the MalGenome dataset (99.15 percent).

**Table 8.** Evaluation metrics of DroidMDetection using different classifiers on the MalGenome dataset.

| Model | Accuracy | Precision | Recall | F1-Score |
| --- | --- | --- | --- | --- |
| AE | 92.95 | 93.60 | 92.20 | 92.90 |
| LSTM | 96.95 | 96.81 | 97.10 | 96.95 |
| CNN | 97.45 | 97.12 | 97.80 | 97.46 |
| LSTM-CNN | 99.15 | 99.00 | 99.30 | 99.15 |

**Table 9.** Evaluation metrics of DroidMDetection using different classifiers on the Drebin dataset.

| Model | Accuracy | Precision | Recall | F1-Score |
| --- | --- | --- | --- | --- |
| AE | 92.60 | 93.29 | 91.80 | 92.54 |
| LSTM | 97.05 | 96.17 | 98.00 | 97.08 |
| CNN | 97.20 | 97.39 | 97.00 | 97.17 |
| LSTM-CNN | 98.65 | 98.50 | 98.80 | 98.65 |

**Table 10.** Evaluation metrics of DroidMDetection using different classifiers on the MalDozer dataset.

| Model | Accuracy | Precision | Recall | F1-Score |
| --- | --- | --- | --- | --- |
| AE | 92.50 | 92.93 | 92.00 | 92.46 |
| LSTM | 96.35 | 96.40 | 96.30 | 96.35 |
| CNN | 97.12 | 97.07 | 97.17 | 97.12 |
| LSTM-CNN | 99.15 | 99.40 | 98.90 | 99.15 |

**Table 11.** Evaluation metrics of DroidMDetection using different classifiers on MaMaDroid dataset.

| Model | Accuracy | Precision | Recall | F1-Score |
| --- | --- | --- | --- | --- |
| AE | 90.71 | 91.63 | 90.37 | 90.99 |
| LSTM | 95.60 | 95.78 | 95.40 | 95.59 |
| CNN | 96.45 | 96.22 | 96.70 | 96.46 |
| CNN-LSTM | 97.50 | 97.88 | 97.10 | 97.49 |

The outcomes of the LSTM, AE, LSTM-CNN and CNN models on the Drebin dataset are displayed in Table 9. The accuracy of the LSTM-CNN model was very high (98.65%). High levels of accuracy (97.20 and 97.05 respectively) were also demonstrated by the LSTM and CNN models, and the AE model's performance was commendable.

The comparison of the LSTM, AE, LSTM-CNN and CNN models on the MalDozer dataset is shown in Table 10. When compared to the LSTM, CNN and AE models on the MalDozer dataset, the CNN-LSTM model scored the highest accuracy (99.15 percent).

Table 11 displays the outcomes of applying the LSTM, AE, LSTM-CNN and CNN models to the MaMaDroid dataset. High precision (97.50%) was achieved with the LSTM-CNN model. Additionally, both the CNN and LSTM models demonstrated impressive accuracy, and the AE model's performance was commendable.

By comparing it to other models, we find that CNN-LSTM achieves a much higher evaluation index for DroidMdetection with feature selection. When using CNN-LSTM, the

F1-score and accuracy of DroidMdetection are improved compared to AE, CNN, and LSTM. Consequently, we settled on the CNN-LSTM classifier as our ultimate recommended model to test and validate the model's generalization skills. DroidMDetection, a CNN-LSTM hybrid classifier, has been shown to have good accuracy and stability in the domain of Android malware detection.

**(2)    Impact of feature selection**

With the use of feature selection, we can narrow down the pool of potential features to those that will have the most impact. To train and test our classifier models, we used the resulting reduced dataset. In the first stage of training, the entire dataset and all the features from the four datasets were employed. The results of the preliminary tests are summarized in Table 11. Table 12 shows that a combined accuracy of 99.15% was achieved on the MalGenome and MalDozer datasets. This finding suggests the great precision required as we advance in developing a more lightweight model. In Table 12, we can see how the proposed model using the CNN-LSTM classifier performed on all four datasets in terms of accuracy, precision, recall, F1 score, and training and testing times. Checking out Table 13, we see that the accuracy is reduced by less than 1% in the Derbin dataset, 0.45% in the MalDozer dataset, and 0.95% in the MalGenome dataset, and the same holds for the MaMaDroid dataset. In all datasets, the timing parameters got better once the feature reduction was applied. A classifier model is an effective tool for malware identification because of its high accuracy and low FN value. The performance results before and after applying the feature selection step in terms of accuracy, recall, precision and f-measure over various datasets is shown in Figures 7–10.

**Table 12.** Performance results when using all features in the dataset with CNN-LSTM classifier.

| Dataset | Accuracy | Precision | Recall | F1-Score | Training Time | Testing Time |
|---|---|---|---|---|---|---|
| Derbin | 98.65 | 98.50 | 98.80 | 98.65 | 0.6267 | 1.3481 |
| MalGenome | 99.15 | 99.00 | 99.30 | 99.19 | 0.5341 | 0.8102 |
| MalDozer | 99.15 | 99.40 | 98.90 | 99.15 | 0.7832 | 3.5542 |
| MaMaDroid | 97.50 | 97.88 | 97.10 | 97.49 | 0.9348 | 2.4531 |

**Table 13.** Performance results when using all features in the dataset with CNN-LSTM classifier.

| Dataset | Accuracy | Precision | Recall | F1-Score | Training Time | Testing Time |
|---|---|---|---|---|---|---|
| Derbin | 97.65 | 97.51 | 97.80 | 97.66 | 0.3214 | 0.5221 |
| MalGenome | 98.20 | 98.01 | 98.40 | 98.20 | 0.0311 | 0.1242 |
| MalDozer | 98.70 | 98.51 | 98.90 | 98.70 | 0.1432 | 0.9312 |
| MaMaDroid | 97.00 | 96.35 | 97.70 | 97.02 | 0.4102 | 1.1021 |

**(3)    Dataset Size Effect**

Table 14 shows that employing CNN-LSTM for detection results in just a little change when the build set percentage is reduced from 90% to 50% of the whole dataset. For this reason, it is important to keep in mind that build_data = {train_data, valid_data} already includes 75% training data and 25% validation data, resulting in a smaller dataset for model training. Even still, DroidMDetection's detection capabilities remain robust in such environments.

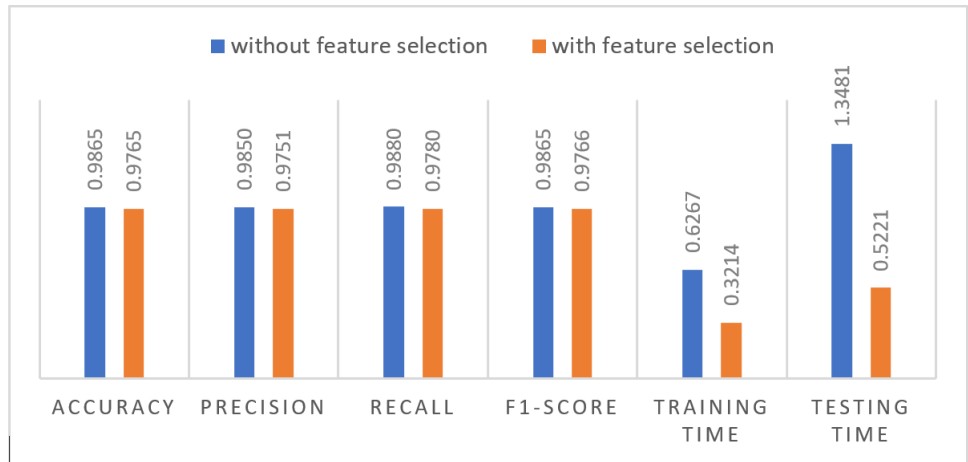

**Figure 7.** Performance measures before and after feature selection with CNN-LSTM on Drebin dataset.

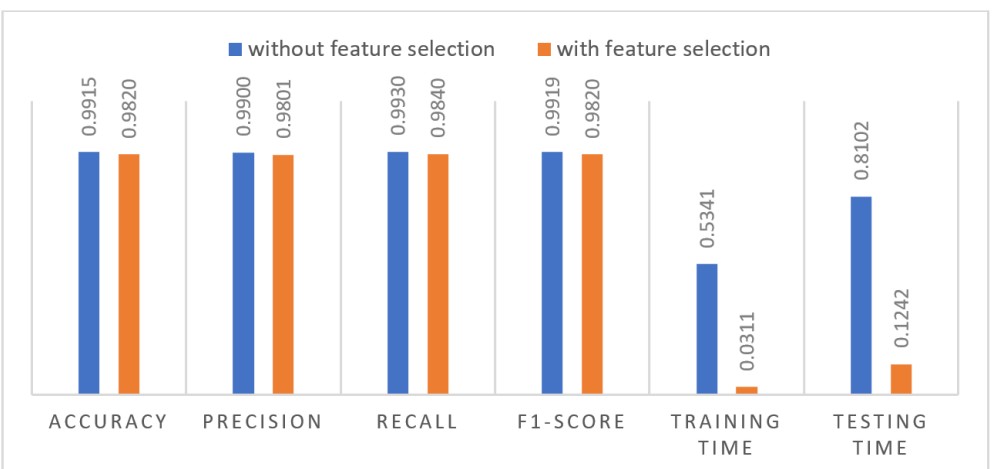

**Figure 8.** Performance measures before and after feature selection with CNN-LSTM on MalGenome dataset.

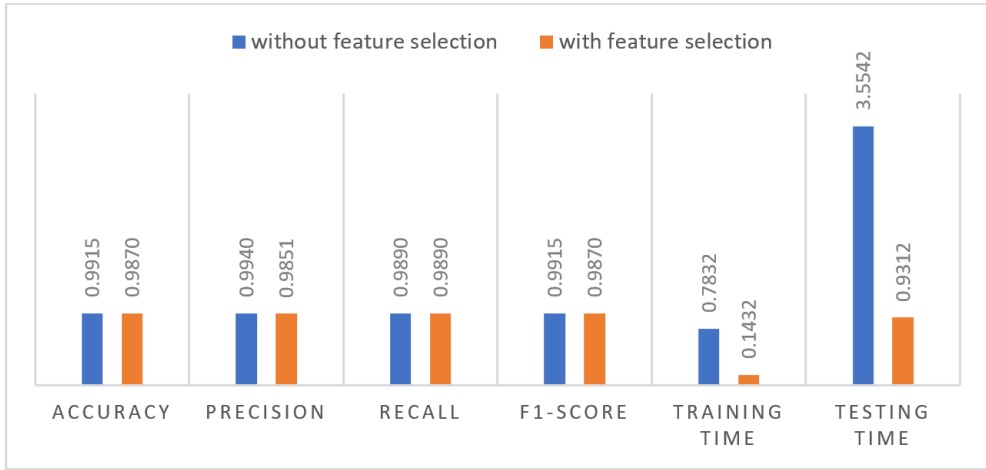

**Figure 9.** Performance measures before and after feature selection with CNN-LSTM on MalDozer dataset.

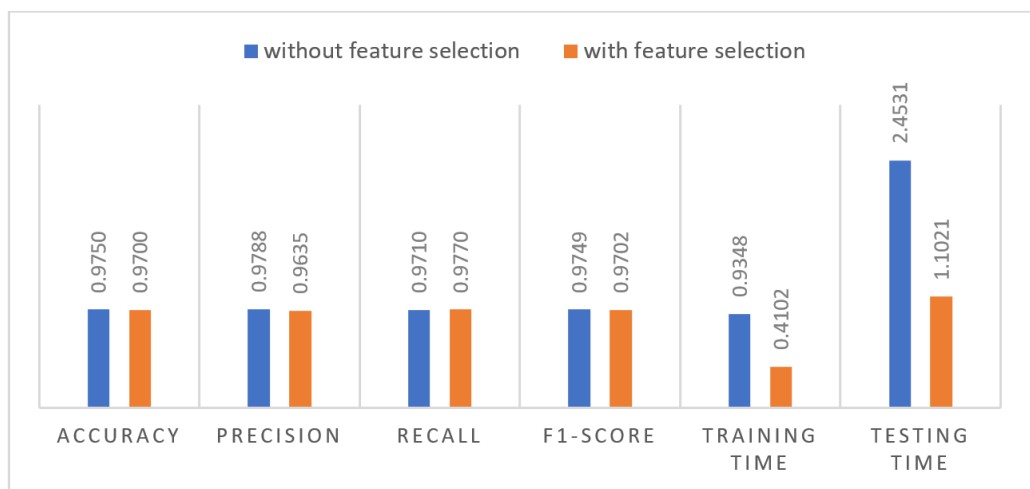

**Figure 10.** Performance measures before and after feature selection with CNN-LSTM on MaMaDroid dataset.

**Table 14.** Performance results when using feature selection in the dataset with CNN-LSTM classifier.

| Dataset | F1-Score | | |
|---|---|---|---|
| | 50% | 70% | 80% |
| Derbin | 96.45 | 97.01 | 98.49 |
| MalGenome | 96.77 | 98.01 | 99.15 |
| MalDozer | 97.28 | 98.51 | 99.25 |
| MAMADROID | 96.52 | 97.27 | 99.05 |

### 4.3.3. Family Clustering

Here, we show off DroidMDetection's family grouping abilities on some purely malware apps (reference datasets). Once a large number of malicious Android apps have been detected by DroidMDetection, the next phase is to cluster them into families. Depending on the setup, the number of identified apps may also change. We evaluate the efficiency of family clustering using the coverage and homogeneity measures. The generated family clusters' integrity is measured by their homogeneity score.

Each generated cluster contains only malware samples from a single family, representing absolute homogeneity. The coverage metrics rate the portion of the clustered dataset that may be trusted. After clustering all of the samples in the dataset using family matching (an optional step), we also give the clustering performance.

Table 15 summarizes the homogeneity and coverage scores for the clustering performance with and without applying the family matching. To begin, DroidMDetection can generate clusters with a high degree of homogeneity (between 91% and 97%) while still providing sufficient coverage (54% on average). The 54% coverage may seem low, but we believe that it is sufficient since (1) increasing it may degrade the quality of the clusters that are generated if we do so. Perfect coverage (with a high error rate) provided by the K-Means clustering algorithm would not always be preferred to high-confidence clusters with adequate coverage. (2) Most malware families in the evaluation datasets only have a small number of samples available. Due to the small size of the discovered dataset, there are rarely more than five samples per malware family, making clustering a challenging task. During deployment, we would be able to include data from sources outside of the clusters in the next round of clustering. It is possible that we could collect enough data to begin identifying patterns among the long-tail malware families. Second, DroidMDetection clusters all the samples in the dataset once the family matching is applied, and the resulting homogeneity drops to 81–89%, an acceptable range.

**Table 15.** The efficiency of family clustering.

| Dataset | DBSCAN Clustering | | After Family Matching | |
|---|---|---|---|---|
| | Coverage | Homogeneity | Coverage | Homogeneity |
| Derbin | 51% | 93.36% | 100% | 83.91% |
| MalGenome | 44% | 91.54% | 100% | 81.32% |
| MAMADROID | 59% | 94.62% | 100% | 85.39% |
| MalDozer | 63% | 97.67% | 100% | 89.12% |

4.3.4. Obfuscation Resiliency

This section details the detection capabilities of DroidMDetection when applied to Android applications that have been obfuscated. First, we try out the DroidChameleon [52] obfuscation tool on a dataset that we have previously created, and second, we use the PRAGuard [40] obfuscation dataset to do our experiments (10k). The PRAGuard experiment combines the PRAGuard dataset with a random selection of AndroZoo's benign Android apps. We created a test dataset that is the same size as the training dataset, which includes both the training and validation datasets. DroidMDetection's detection efficacy against various obfuscation methods is shown in Table 16. DroidMDetection's nearly average detection rate F1-score is 99.61%, which demonstrates its excellent resistance to prevalent obfuscation methods.

**Table 16.** DroidMDetection obfuscation resistance on PRAGuard data.

| Obfuscation Techniques | Accuracy | Precision | Recall | F1-Score |
|---|---|---|---|---|
| String Encryption | 99.45 | 99.50 | 99.40 | 99.45 |
| Trivial | 99.40 | 99.20 | 99.60 | 99.40 |
| Class Encryption | 99.60 | 99.50 | 99.70 | 99.60 |
| Reflection | 99.20 | 99.10 | 99.30 | 99.20 |
| (1) + (2) | 99.45 | 99.60 | 99.30 | 99.45 |
| (1) + (2) +(3) | 99.65 | 99.60 | 99.70 | 99.65 |
| (1) + (2) +(3) +(4) | 99.65 | 99.80 | 99.50 | 99.65 |

In Table 17, we compare DroidMDetection to various other obfuscation methods in the DroidChameleaon experiment. Samples of both benign code randomly selected from AndroZoo and malicious code were masked and included in the created dataset originally from Drebin. To simplify the LSTM-CNN training process, we merely train using one obfuscation method (Table 17) and then evaluate the other methods. The results of obfuscation resilience on a dataset created by DroidChameleon are shown in Table 16. DroidMDetection's results demonstrate its reliability. This experiment shows that DroidMDetection, when trained on non-obfuscated datasets, can identify malware that has been obfuscated using popular techniques. We attribute DroidMDetection's resistance to obfuscation to the fact that Android API sequences are used as features in ML development. A fundamental part of any Android app is its use of Android APIs. A malware author cannot hide API access if the malicious payload is not downloaded at runtime. So long as they do not delete or hide API access calls, the most common types of obfuscation can be tolerated by DroidMDetection.

**Table 17.** DroidMDetection obfuscation resistant on DroidChameleon dataset.

| Obfuscation Techniques | Accuracy | Precision | Recall | F1-Score |
|---|---|---|---|---|
| Method Renaming | 99.60 | 99.70 | 99.50 | 99.60 |
| Class Renaming | 99.50 | 99.70 | 99.30 | 99.50 |
| String Encryption | 99.75 | 99.70 | 99.80 | 99.75 |
| Field Renaming | 99.75 | 99.90 | 99.60 | 99.75 |
| Call Indirection | 99.60 | 99.40 | 99.80 | 99.60 |
| Array Encryption | 99.40 | 99.40 | 99.40 | 99.40 |
| Junk Code Insertion | 99.25 | 99.30 | 99.20 | 99.25 |
| Code Reordering | 99.15 | 99.30 | 99.00 | 99.15 |
| Debug Information Removing | 99.50 | 99.30 | 99.70 | 99.50 |
| Instruction Insertion | 99.60 | 99.50 | 99.70 | 99.60 |
| Disassembling and Reassembling | 99.60 | 99.80 | 99.40 | 99.60 |

### 4.3.5. Time Efficiency

We provide the typical detection time using DroidMDetection here. Disassembly, preparation, and inference time are all part of the detection process. On average, Droid-MDetection takes 3.8 s to generate a fingerprint for an Android app. Due to higher package sizes, benign programs take 5.8 s longer to launch than malicious ones. DroidMDetection's average fingerprinting time for malware apps is 3.5 s.

### 4.3.6. Comparative Study

Here, we examine the similarities and differences between DroidMDetection and four other cutting-edge Android malware detection systems: DroidMalwareDetector, DroidAPIMiner, MaMaDroid, and MalDozer. To evaluate how DroidMDetection stacks up against other related work, we used an identical dataset like the one used in MaMaDroid (malicious and benign apps) and evaluation settings provided by the authors in [46]. This collection contains 35.5K malicious apps in addition to 8.5K benign ones from the Drebin dataset. According to MaMaDroid's most recent assessment, malicious applications from 012 (Drebin), 2013, 2014, 2015, and 2016 are classified as newbenign and oldbenign. In Table 18, we see how MaMaDroid, DroidAPIMiner, DroidMalwareDetector, MalDozer, and DroidMDetection fare when pitted against one another across a variety of datasets. The results of DroidMDetection are shown to the user in the form of an F1-score.

**Table 18.** Detection performance of DroidMalwareDetector, MaMaDroid, DroidMDetection, and DroidAPIMiner.

| Dataset | F1-Score | | | | |
|---|---|---|---|---|---|
| | Proposed | DroidAPIMiner | MaMaDroid | MalDozer | DroidMalwareDetector |
| 2016 & newbenign | 98.52 | 36.00 | 92.00 | 91.13 | 95.22 |
| 2015 & newbenign | 97.32 | 77.00 | 95.00 | 94.31 | 96.22 |
| 2014 & newbenign | 99.04 | 92.00 | 99.00 | 93.22 | 98.43 |
| 2014 & oldbenign | 99.20 | 62.00 | 95.00 | 94.45 | 98.24 |
| 2013 & oldbenign | 98.16 | 36.00 | 97.00 | 89.23 | 96.50 |
| drebin & oldbenign | 99.05 | 32.00 | 96.00 | 91.61 | 97.14 |

Table 18 shows that across the board, DroidMDetection is superior to MaMaDroid, DroidAPIMiner, DroidMalwareDetector, and MalDozer. Tables 19–26 also provide a comparison of the proposed system's performance to that of existing state-of-the-art methods. The performance under varying dataset conditions is shown in Tables 19–26 which include training on an outdated malware dataset and testing on a more recent one. In most situations, DroidMDetection works better than (or at least gets very similar results to) alternative methods. The performance measurements in Tables 19–26 reveal that the suggested system is superior to most state-of-the-art methods. This means that even Android smartphones

with less processing power can benefit from our proposed system. The proposed technique effectively reduced the size of the datasets without sacrificing accuracy. This lays the groundwork for future studies to use the reduced dataset to create more nimble and effective malware detection algorithms. The proposed methodology is highly effective and efficient, uses minimal resources, and can be applied to a wide variety of classification problems while still maintaining its high level of accuracy and efficiency. The time savings were achieved throughout the process of selecting features. These impressive measures of performance demonstrate that the created classifier can generalize to data beyond the training set.

**Table 19.** Performance measures of proposed framework and related work.

| | Drebin & Oldbenign | | | 2013 & Oldbenign | | | 2014 & Oldbenign | | |
|---|---|---|---|---|---|---|---|---|---|
| | Miner | Doser | Proposed | Miner | Dozer | Proposed | Miner | Doser | Proposed |
| drebin & oldbenign | 32.0% | 91.2% | 99.3% | 35.0% | 92.4% | 98.3% | 34.0% | 88.4% | 99.2% |
| 2013 & oldbenign | 33.0% | 93.5% | 98.2% | 36.0% | 91.4% | 98.3% | 35.0% | 83.4% | 96.4% |
| 2014 & oldbenign | 36.0% | 90.0% | 99.1% | 39.0% | 83.4% | 89.4% | 62.0% | 66.2% | 98.5% |

**Table 20.** Performance measures of proposed framework and related work.

| | 2015 & Oldbenign | | | 2016 & Oldbenign | | |
|---|---|---|---|---|---|---|
| | Miner | Doser | Proposed | Miner | Dozer | Proposed |
| drebin & oldbenign | 30.0% | 53.4% | 89.3% | 33.0% | 45.6% | 47.1% |
| 2013 & oldbenign | 31.0% | 67.4% | 90.1% | 33.0% | 88.3% | 80.2% |
| 2014 & oldbenign | 33.0% | 89.3% | 91.4% | 37.0% | 74.2% | 77.1% |

**Table 21.** Performance measures of proposed framework and related work.

| | Drebin & Newbenign | | | 2013 & Newbenign | | | 2014 & Newbenign | | |
|---|---|---|---|---|---|---|---|---|---|
| | Miner | Doser | Proposed | Miner | Dozer | Proposed | Miner | Doser | Proposed |
| 2014 & newbenign | 76.0% | 88.2% | 98.6% | 75.0% | 92.4% | 99.5% | 92.0% | 90.2% | 99.3% |
| 2015 & newbenign | 68.0% | 91.4% | 99.4% | 68.0% | 87.4% | 98.1% | 69.0% | 87.1% | 95.1% |
| 2016 & newbenign | 33.0% | 90.4% | 99.6% | 35.0% | 85.1% | 98.2% | 36.0% | 67.3% | 88.4% |

**Table 22.** Performance measures of proposed framework and related work.

| | 2015 & Newbenign | | | 2016 & Newbenign | | |
|---|---|---|---|---|---|---|
| | Miner | Doser | Proposed | Miner | Dozer | Proposed |
| 2014 & newbenign | 67.0% | 84.1% | 93.2% | 65.0% | 92.4% | 95.4% |
| 2015 & newbenign | 77.0% | 91.3% | 96.2% | 65.0% | 47.1% | 91.3% |
| 2016 & newbenign | 34.0% | 90.1% | 98.2% | 36.0% | 83.4% | 92.2% |

**Table 23.** Performance measures of proposed framework and related work.

| | Drebin & Oldbenign | | | 2013 & Oldbenign | | | 2014 & Oldbenign | | |
|---|---|---|---|---|---|---|---|---|---|
| | MaMa | MD | Proposed | MaMa | MD | Proposed | MaMa | MD | Proposed |
| drebin & oldbenign | 96.0% | 97.3% | 99.3% | 95.0% | 95.2% | 98.3% | 72.0% | 92.1% | 99.2% |
| 2013 & oldbenign | 94.0% | 93.2% | 98.2% | 97.0% | 97.2% | 98.3% | 73.0% | 90.2% | 96.4% |
| 2014 & oldbenign | 92.0% | 94.0% | 99.1% | 93.0% | 90.2% | 89.4% | 95.0% | 97.1% | 98.5% |

**Table 24.** Performance measures of proposed framework and related work.

| | 2015 & Oldbenign | | | 2016 & Oldbenign | | |
|---|---|---|---|---|---|---|
| | MaMa | MD | Proposed | MaMa | MD | Proposed |
| drebin & oldbenign | 39.0% | 88.0% | 89.3% | 42.0% | 61.2% | 47.1% |
| 2013 & oldbenign | 37.0% | 62.1% | 90.1% | 28.0% | 42.0% | 80.2% |
| 2014 & oldbenign | 78.0% | 85.1% | 91.4% | 37.0% | 55.0% | 77.1% |

**Table 25.** Performance measures of proposed framework and related work.

| | Drebin & Newbenign | | | 2013 & Newbenign | | | 2014 & Newbenign | | |
|---|---|---|---|---|---|---|---|---|---|
| | MaMa | MD | Proposed | MaMa | MD | Proposed | MaMa | MD | Proposed |
| 2014 & newbenign | 98.0% | 98.1% | 98.6% | 98.0% | 98.2% | 99.5% | 99.0% | 99.0% | 99.3% |
| 2015 & newbenign | 97.0% | 97.2% | 99.4% | 97.0% | 96.0% | 98.1% | 99.0% | 98.2% | 95.1% |
| 2016 & newbenign | 96.0% | 98.0% | 99.6% | 98.0% | 88.1% | 98.2% | 98.0% | 98.1% | 98.4% |

**Table 26.** Performance measures of proposed framework and related work.

| | 2015 & Newbenign | | | 2016 & Newbenign | | |
|---|---|---|---|---|---|---|
| | MaMa | MD | Proposed | MaMa | MD | Proposed |
| 2014 & newbenign | 85.0% | 88.2% | 93.2% | 81.0% | 87.0% | 95.4% |
| 2015 & newbenign | 95.0% | 96.0% | 96.2% | 88.0% | 72.4% | 91.3% |
| 2016 & newbenign | 92.0% | 94.0% | 98.2% | 92.0% | 91.5% | 92.2% |

*4.4. Reliability in the Face of Evolving Threats*

To address the concern regarding the impact of earlier studies on the reliability of the proposed approach in the face of evolving malware and benign apps, we have further analyzed and evaluated the potential implications. It is crucial to consider the dynamic nature of the malware landscape and its continuous evolution, which can potentially lead to the degeneration of detection models over time. In our study, we recognize the importance of regularly updating and adapting the proposed model to effectively combat evolving threats. By closely monitoring the changing characteristics of malware and benign apps, we can identify patterns and trends that may impact the performance of the detection framework. This ongoing analysis enables us to refine the model and incorporate relevant adjustments to maintain its effectiveness.

In this section, we discuss the challenges posed by the dynamic nature of the malware ecosystem and the measures taken to mitigate the potential degradation of the proposed model. We discuss the potential implications of the evolving malware landscape and the measures taken to address them within the proposed approach.

1. Understanding the Evolution of Malware and Benign Apps: We provide an overview of the dynamic nature of malware and benign apps, highlighting the rapid evolution, polymorphic behavior, and obfuscation techniques employed by malicious actors. This understanding is crucial to comprehend the challenges posed to malware detection and the potential impact on the reliability of the proposed model.
2. Challenges in Maintaining Model Effectiveness: We acknowledge that the continuous evolution of malware and benign apps can introduce new variants, making it necessary to adapt the detection model to capture emerging threats effectively. We discuss the potential consequences of not addressing these challenges, such as false negatives, decreased detection accuracy, and increased vulnerability to new attack vectors.
3. Adaptive and Continuous Learning Approaches: To mitigate the effects of evolving threats, we employ adaptive and continuous learning techniques within the proposed framework. These approaches allow the model to dynamically update its knowledge and adapt to changing patterns and characteristics of malware and benign apps. We

4. explain the strategies used, such as incremental learning, ensemble methods, and regular model retraining, to ensure the model remains up to date and effective.

4. Collaborative Intelligence and Threat Intelligence Integration: Recognizing the importance of collective efforts, we highlight the integration of collaborative intelligence and threat intelligence sources in the proposed approach. By leveraging real-time information on emerging threats, malware signatures, and behavioral patterns, the model can enhance its detection capabilities and adapt to the evolving threat landscape.

5. Evaluation of Model Robustness: We provide insights into the evaluation of the proposed model's robustness in the face of evolving threats. This includes benchmarking against evolving malware datasets, measuring the detection rate over time, and assessing the model's ability to identify new malware variants and benign app changes. The evaluation demonstrates the model's resilience and ability to adapt to the dynamic nature of the ecosystem.

We provide a comprehensive understanding of the proposed approach's response to the challenges posed by the continuous evolution of malware and benign apps. This analysis emphasizes the proactive measures taken to maintain the reliability and effectiveness of the model, ensuring its practicality in real-world scenarios.

*4.5. Sustainability and Resilience against Evolution*

In order to address the challenge of sustainability and resilience against the rapid evolution of malware and benign apps, it is crucial to consider the long-term effectiveness of the proposed model. While the performance of the constructed classifier has been demonstrated on the evaluated apps, it is essential to assess its adaptability to future apps that may exhibit novel behaviors and evasion techniques. To ensure the reliability and continuous efficacy of the model, proactive measures need to be taken to mitigate the impact of evolving features. This involves monitoring the changing landscape of malware and benign apps, identifying indicators of deteriorating features, and implementing timely updates to the model.

We present strategies and considerations employed in our approach to enhance sustainability. This includes adaptive feature selection mechanisms, continuous model retraining with evolving datasets, and leveraging ensemble techniques to combine multiple models trained on different time periods. These strategies aim to address the challenge of evolving apps and ensure the sustainability of the proposed approach. We describe the evaluation methodology used to assess the sustainability and resilience of the proposed approach. This includes conducting experiments and analyses to measure the model's performance over time, tracking the evolution of features and their impact on detection accuracy, and examining the model's adaptability to new app variants. The results of these evaluations provide insights into the sustainability aspect of our approach.

**5. Conclusions**

Recently, the mobile ecosystem has faced a serious security threat from mobile malware. To solve security concerns, which are typically concerned with the effective performance of the chosen classifiers as well as the impactful selection of features, deep learning algorithms needed to be more accurate. To demonstrate that it is possible to increase accuracy by reducing the number of permissions while retaining high efficiency and effectiveness, we statically examined the Android ecosystem. We introduced DroidMDetecion, an NLP-based deep learning approach to Android malware detection. In this study, the primary static feature types of Android apps were first extracted, and word embeddings were used to characterize them. Then, we use feature selection to narrow the number of features to the most pertinent subset. The classifier was created using deep learning using CNN-LSTM. To cluster extremely similar malicious programs into their most likely malware family groups, DBScan clustering is added on top of FeVec and deep auto-encoder capabilities. We assess it using a variety of real-world datasets of both benign and malicious apps. According to experimental findings, DroidMDetecion outperforms some harmful detection programs in

terms of accuracy and execution efficiency. The current work is limited to android apps, but not all phones are on android apps besides the large extension of iOS systems. This study could be expanded upon to encompass other operating systems, such as iOS, in future work. Then, to create a new dataset that we would label, new tools for extracting static features should be created. Additionally, all findings concerning learning methodologies, evaluation metrics, and hyperparameter settings could be used for the training of neural networks. It would be important to update the dataset with the most recent labelling methods for more research and to create an automated tool for an automatically updating neural network.

**Author Contributions:** Conceptualization, F.T.; methodology, O.A.F.; software, H.A.H.; validation, H.A.H. and F.T.; formal analysis, M.A.K.; investigation, O.A.F.; resources, F.T.; data curation, F.T.; writing—original draft preparation, S.A.; writing—review and editing, S.A.; visualization, H.A.H.; supervision, F.T.; project administration, F.T.; funding acquisition, F.T. All authors have read and agreed to the published version of the manuscript.

**Funding:** UAE University and Zayed University joint research grant.

**Institutional Review Board Statement:** Not applicable.

**Informed Consent Statement:** Not applicable.

**Data Availability Statement:** Our evaluation dataset includes millions of Android apps gathered over the past decade to serve as a sampling space for our studies. Our study is convincing because of the breadth of size, time, and malware families that it covers. Malware from Drebin, MalGenome, MalDozer, and MaMaDroid were used to test DroidMDetection's family clustering and detection capabilities. In addition, we employed benign programs from the AndroZoo repository.

**Conflicts of Interest:** The authors declare no conflict of interest.

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
