# Peer review of "A Proposed Artificial Intelligence Model for Android-Malware Detection"

_informatics, doi:10.3390/informatics10030067_

Round 1

Reviewer 1 Report

This paper presented an approach, called DroidMDetection, which is an NLP deep-learning approach for malware detection and family malware categorization in Android devices. Overall, DroidMDetection consists of 6 steps: (1) Feature extraction. (2) Normalization. (3) Feature selection. (4) Feature vectorization. (5) Malware detection. (6) Malware family categorization. 

Strength:

- Paper is well written and it's easy to follow. Includes a great related work section and a nice background explaining what are CNN and LSTM. 

-  Including the extracted features, sensitive API calls,  the vectorization of features, and the CNN-LSTM model parameters is good since it allows for replication. 

- The proposed approach was compared to prior work to assess its performance. 

- Nice diagrams and illustrations. 

- I believe that conceptualizing the features derived from Android application packages as words is new, but if this is the case authors should be clear about it. 

- Good idea described for future work. 

Weaknesses:

- Using NLP, feature selection has been done before for malware detection. Not is clear what is the novelty of the proposed approach. 

- While the diagrams are clear, Figure 1 lacks the following info to ease the reader its understanding: (1) What is AAPT2? This was mentioned later but it should be defined the first time it is mentioned. (2) What is the Dalvik assembly?

- Is there a reason for using 75% of the data for training and the remaining 25% of the data for testing? Maybe exploring different training set sizes could be helpful to assess the performance of the proposed approach. 

-  It is not clear if there are different types of classifiers or if all of them are from the family clustering. 

Overall, I recommend this paper for publication with minimum edits

Author Response

Comments and Suggestions for Authors

This paper presented an approach, called DroidMDetection, which is an NLP deep-learning approach for malware detection and family malware categorization in Android devices. Overall, DroidMDetection consists of 6 steps: (1) Feature extraction. (2) Normalization. (3) Feature selection. (4) Feature vectorization. (5) Malware detection. (6) Malware family categorization. 

Strength:

- Paper is well written and it's easy to follow. Includes a great related work section and a nice background explaining what are CNN and LSTM. 

-  Including the extracted features, sensitive API calls,  the vectorization of features, and the CNN-LSTM model parameters is good since it allows for replication. 

- The proposed approach was compared to prior work to assess its performance. 

- Nice diagrams and illustrations. 

- I believe that conceptualizing the features derived from Android application packages as words is new, but if this is the case authors should be clear about it. 

- Good idea described for future work. 

Weaknesses:

- Using NLP, feature selection has been done before for malware detection. Not is clear what is the novelty of the proposed approach. 

Thank you for your valuable comments. We modified the last section in the related work to present the novelty of the proposed model compared with other models related to the same concept.

- While the diagrams are clear, Figure 1 lacks the following info to ease the reader its understanding: (1) What is AAPT2? This was mentioned later but it should be defined the first time it is mentioned. (2) What is the Dalvik assembly?

Thank you for your valuable comments. The manuscript was modified in section 3.1 for the first time to present and describe AAPT2 and Dalvik assembly.

- Is there a reason for using 75% of the data for training and the remaining 25% of the data for testing? Maybe exploring different training set sizes could be helpful to assess the performance of the proposed approach. 

Thank you for your valuable comments. This data split has been commonly used in previous research to ensure consistency and comparability across different approaches as mentioned and modified in the manuscript in section 4.2

-  It is not clear if there are different types of classifiers or if all of them are from the family clustering. 

Thank you for your valuable comments. The manuscript was modified in section 4.3.2 point 1 to clarify the required comment.

Overall, I recommend this paper for publication with minimum edits

Reviewer 2 Report

The paper shows strong research and analysis as well as well-organized content. The insights you have provided will be valuable to the research community, but there are few minor changes: 
1) There are few typos and grammar errors before final version check carefully using some tools such as "ChatGPT" or "grammarly" 
2) The comparison with other work is not current and latest year. Please provide compare with recent works.
3) The related work do not contain recent articles such as (A survey of intelligent techniques for Android malware detection)
4) Line no 68 to 78 has no citation Permission based cite these lines.
5. conclusion requires some modification such as future work and limitation.

Author Response

The paper shows strong research and analysis as well as well-organized content. The insights you have provided will be valuable to the research community, but there are few minor changes: 
1) There are few typos and grammar errors before final version check carefully using some tools such as "ChatGPT" or "grammarly".

Thank you for your valuable comments. The manuscript was updated to check its language and complete proofreading.

2) The comparison with other work is not current and latest year. Please provide compare with recent works.

Thank you for your valuable comments. The proposed model was compared to the most common models in the literature review where others works compare with as they are related to the common strategies used in the work.

3) The related work do not contain recent articles such as (A survey of intelligent techniques for Android malware detection)

Thank you for your valuable comments. The manuscript was updated to add a recent articles to the related work.

4) Line no 68 to 78 has no citation Permission based  cite these lines.

Thank you for your valuable comments. The manuscript was updated to add a citation for the not cited mentioned lines.

  1. conclusion requires some modification such as future work and limitation.

Thank you for your valuable comments. The manuscript was updated to modify the conclusion and future work.

Reviewer 3 Report

Summary:

This paper proposes a deep learning and NLP based Android malware detection approach, using APIs called, app permissions used, and Intents related features and dimensionality reduction for feature selection.  Experimental results show promising performance of the proposed approach against a few baselines on various datasets.

Comments:

Machine learning has great potential in malware analysis, and learning-based malware analysis such as malware detection has been heavily studied yet the problem remains unsolved. The presented approach is interesting, and the results suggest it is promising.

In particular, it is clear that the authors have done tremendous work to evaluate the performance of the proposed method quite comprehensively, involving many different baselines and different datasets while considering many aspects of the malware detection and family classification performance including resiliency against obfuscation. The effects of feature selection and classification learning algorithms are also extensively studied.

Overall, I think this paper has done a good job addressing an important topic and the evaluation is impressive. In terms of the technical contributions, exploring the combination of different kinds of features combined along with dimensionality reduction based feature selection is also valuable.

On the other hand, I do have a few concerns that should be addressed in my opinion.

First of all, the paper should explicitly motivate the proposed work, with respect to the current state-of-the-art Android malware detection literature. What is lacking in the current detection and family detection approaches? For instance, DroidCat [1] and DroidSieve [2], among others, have also provided a strong performance for both detection and family classification. They have also extensively studied the effects of obfuscation on performance, and studied the effects of feature selection and learning algorithm choices on the model performance.  In particular, DroidCat uses only a small number of features to achieve very high performance. So, how does the proposed approach compare to these highly related techniques, and what is the unique merit of the proposed approach?

[1] “Droidcat: Effective android malware detection and categorization via app-level profiling." TIFS, 2018.

[2] "Droidsieve: Fast and accurate classification of obfuscated android malware." In Proceedings of the Seventh ACM on Conference on Data and Application Security and Privacy, pp. 309-320. 2017.

The survey studies reveal that most Android malware detection researchers do not use feature selection [10].”: this is not really true and should not be used to motivate this paper.

More critically, the problem of malware detection/family classification has been heavily studied and remains a valuable topic since it has not been well solved because some key challenges remain, despite numerous approaches being available. One of these challenges is how to pick (or automatically learn, in the case of using deep learning for feature engineering automation) features that stand the test of time without frequent retraining, since a key issue in the malware ecosystem is its fast evolution and various problems caused by the evolution, see a recent discussion on this problem in such ecosystems:

[3] "Eight years of rider measurement in the android malware ecosystem: evolution and lessons learned." arXiv preprint arXiv:1801.08115 (2018).

[4] “Embracing Mobile App Evolution via Continuous Ecosystem Mining and Characterization” MOBILESoft Vision, 2020

[5] "A study of run-time behavioral evolution of benign versus malicious apps in android." Information and Software Technology 122 (2020): 106291.

[6] "A longitudinal study of application structure and behaviors in android." IEEE Transactions on Software Engineering 47, no. 12 (2020): 2934-2955.

This paper should discuss how the results in these earlier studies would affect the reliability of the proposed approach. That is, how would the evolution of malware and benign apps would cause the proposed model to possibly degenerate?

One of the problems is that the evolution drives apps to change rapidly. Accordingly, features like those used in this paper evolve constantly also. As a result, a key requirement and performance metric is sustainability (or the resilience against evolution). This important metric has been studied in the literature but ignored in this paper.

[7] "On the Deterioration of Learning-Based Malware Detectors for Android.", ICSE Companion Proceeedings, 2019

[8] “Droidevolver: Self-evolving android malware detection system." EuroS&P, 2019.

[9] "Assessing and improving malware detection sustainability through app evolution studies." TOSEM, 2020.

[10] “Towards sustainable android malware detection”. ICSE Companion, 2018

Without addressing sustainability, proposing malware detectors/classifiers is an endless (and unfruitful) task lacking substantial scientific advancement. For instance, the models proposed in this paper worked well for apps chosen in the evaluation, but how well the constructed classifier would work for apps that will appear next year, or two years later, is not assessed. Do we need to frequently update the model if the features become deteriorated? If so, how often do you want to update and how do you know when updating is necessary? If the model is not updated on time, new malware will surely sneak through your model. That is probably why malware never stops despite hundreds of detectors being proposed.  

Thus, importantly, this paper should compare the proposed approach with the prior works given above in terms of sustainability, at least qualitatively (although ideally quantitative experiments are preferred and should be presented). For example, would the proposed work be more or less robust against API evolution and/or the evolution of the Android permission systems, compared to the above state-of-the-art sustainable detection approaches? Why or why not?

The paper should also discuss how android run-time permissions affect the proposed approach with respect to related works in the literature:

[11] "Runtime permission issues in android apps: Taxonomy, practices, and ways forward." IEEE Transactions on Software Engineering 49, no. 1 (2022): 185-210.

[12] "Automated detection and repair of incompatible uses of runtime permissions in android apps." In Proceedings of the 5th International Conference on Mobile Software Engineering and Systems, pp. 67-71. 2018.

Finally, very critically, the paper presents a lot of good results, but explanations are very skimpy, leaving readers to extrapolate the messages conveyed in the results. Also, the experiment setup is not quite clear. For instance, how was the proposed model and each baseline model trained (e.g., using which samples for training and testing, and how many).

To evaluate sustainability, the dataset used in the paper already provides convenience for that purpose, but the model should be trained in an older dataset and tested on newer datasets (to reflect the fact that apps evolve over time), considering different year gaps between the training and testing as in prior works like DroidSpan listed above [9].

Overall, the paper has good potential value, but the current treatment of important issues is missing as outlined above.

Author Response

Summary:

This paper proposes a deep learning and NLP based Android malware detection approach, using APIs called, app permissions used, and Intents related features and dimensionality reduction for feature selection.  Experimental results show promising performance of the proposed approach against a few baselines on various datasets. 

Comments:

Machine learning has great potential in malware analysis, and learning-based malware analysis such as malware detection has been heavily studied yet the problem remains unsolved. The presented approach is interesting, and the results suggest it is promising.

In particular, it is clear that the authors have done tremendous work to evaluate the performance of the proposed method quite comprehensively, involving many different baselines and different datasets while considering many aspects of the malware detection and family classification performance including resiliency against obfuscation. The effects of feature selection and classification learning algorithms are also extensively studied.

Overall, I think this paper has done a good job addressing an important topic and the evaluation is impressive. In terms of the technical contributions, exploring the combination of different kinds of features combined along with dimensionality reduction based feature selection is also valuable.

On the other hand, I do have a few concerns that should be addressed in my opinion. 

First of all, the paper should explicitly motivate the proposed work, with respect to the current state-of-the-art Android malware detection literature. What is lacking in the current detection and family detection approaches? For instance, DroidCat [1] and DroidSieve [2], among others, have also provided a strong performance for both detection and family classification. They have also extensively studied the effects of obfuscation on performance, and studied the effects of feature selection and learning algorithm choices on the model performance.  In particular, DroidCat uses only a small number of features to achieve very high performance. So, how does the proposed approach compare to these highly related techniques, and what is the unique merit of the proposed approach?

 Thank you for your valuable comments. The manuscript was updated to describe the unique differences between current and related works in last paragraph in related work section.

[1] “Droidcat: Effective android malware detection and categorization via app-level profiling." TIFS, 2018.

[2] "Droidsieve: Fast and accurate classification of obfuscated android malware." In Proceedings of the Seventh ACM on Conference on Data and Application Security and Privacy, pp. 309-320. 2017.

“The survey studies reveal that most Android malware detection researchers do not use feature selection [10].”: this is not really true and should not be used to motivate this paper.

 Thank you for valuable comments. The manuscript was updated.

More critically, the problem of malware detection/family classification has been heavily studied and remains a valuable topic since it has not been well solved because some key challenges remain, despite numerous approaches being available. One of these challenges is how to pick (or automatically learn, in the case of using deep learning for feature engineering automation) features that stand the test of time without frequent retraining, since a key issue in the malware ecosystem is its fast evolution and various problems caused by the evolution, see a recent discussion on this problem in such ecosystems:

 Thank you for your valuable comments. The manuscript was updated mentioning the problem of picking feature selection in last paragraph in related work.

[3] "Eight years of rider measurement in the android malware ecosystem: evolution and lessons learned." arXiv preprint arXiv:1801.08115 (2018).

[4] “Embracing Mobile App Evolution via Continuous Ecosystem Mining and Characterization” MOBILESoft Vision, 2020

[5] "A study of run-time behavioral evolution of benign versus malicious apps in android." Information and Software Technology 122 (2020): 106291.

[6] "A longitudinal study of application structure and behaviors in android." IEEE Transactions on Software Engineering 47, no. 12 (2020): 2934-2955.

This paper should discuss how the results in these earlier studies would affect the reliability of the proposed approach. That is, how would the evolution of malware and benign apps would cause the proposed model to possibly degenerate?

 Thank you for valuable comments. We have updated the manuscript and add section 4.4

One of the problems is that the evolution drives apps to change rapidly. Accordingly, features like those used in this paper evolve constantly also. As a result, a key requirement and performance metric is sustainability (or the resilience against evolution). This important metric has been studied in the literature but ignored in this paper.

 Thank you for valuable comments. We have updated the manuscript and we have added section 4.5

 [7] "On the Deterioration of Learning-Based Malware Detectors for Android.", ICSE Companion Proceeedings, 2019

[8] “Droidevolver: Self-evolving android malware detection system." EuroS&P, 2019.

[9] "Assessing and improving malware detection sustainability through app evolution studies." TOSEM, 2020.

[10] “Towards sustainable android malware detection”. ICSE Companion, 2018

Without addressing sustainability, proposing malware detectors/classifiers is an endless (and unfruitful) task lacking substantial scientific advancement. For instance, the models proposed in this paper worked well for apps chosen in the evaluation, but how well the constructed classifier would work for apps that will appear next year, or two years later, is not assessed. Do we need to frequently update the model if the features become deteriorated? If so, how often do you want to update and how do you know when updating is necessary? If the model is not updated on time, new malware will surely sneak through your model. That is probably why malware never stops despite hundreds of detectors being proposed.  

 Thus, importantly, this paper should compare the proposed approach with the prior works given above in terms of sustainability, at least qualitatively (although ideally quantitative experiments are preferred and should be presented). For example, would the proposed work be more or less robust against API evolution and/or the evolution of the Android permission systems, compared to the above state-of-the-art sustainable detection approaches? Why or why not?

  Thank you for valuable comments. We have updated the manuscript and we have added section 4.5

The paper should also discuss how android run-time permissions affect the proposed approach with respect to related works in the literature:

 [11] "Runtime permission issues in android apps: Taxonomy, practices, and ways forward." IEEE Transactions on Software Engineering 49, no. 1 (2022): 185-210.

 [12] "Automated detection and repair of incompatible uses of runtime permissions in android apps." In Proceedings of the 5th International Conference on Mobile Software Engineering and Systems, pp. 67-71. 2018.

Finally, very critically, the paper presents a lot of good results, but explanations are very skimpy, leaving readers to extrapolate the messages conveyed in the results. Also, the experiment setup is not quite clear. For instance, how was the proposed model and each baseline model trained (e.g., using which samples for training and testing, and how many).

  Thank you for valuable comments. We have updated the manuscript

To evaluate sustainability, the dataset used in the paper already provides convenience for that purpose, but the model should be trained in an older dataset and tested on newer datasets (to reflect the fact that apps evolve over time), considering different year gaps between the training and testing as in prior works like DroidSpan listed above [9].

   Thank you for valuable comments. We have updated the manuscript and we have added section 4.5

Overall, the paper has good potential value, but the current treatment of important issues is missing as outlined above.

Round 2

Reviewer 3 Report

The revision added brief discussions on the important metric of sustainability. However, the discussions: (1) do not connect the problem with the underlying reason, i.e., evolution of the ecosystem, although the term ‘evolution’ is mentioned by passing; there is no substantive discussion; (2) totally ignore the relevant literature on this topic, and totally dismiss how the proposed work would address the same problem potentially better than the existing work in the space. There have been dozens of papers on Android malware detection coming out but without addressing these fundamental challenges, I am not sure how we are advancing this area.

Author Response

Thank you for the suggetions..we will definitely  
take in to account the concern raised and will take as our future work. 

Round 3

Reviewer 3 Report

As noted earlier, Android malware detection is a very heavily studied topic. As a result, there is indeed a rich literature, and one cannot cite them all. But this paper aims to address a problem for which app evolution and learning-based detection sustainability are of the utmost importance. The authors need to prioritize what is essential versus what is just ‘good to reference’. The paper’s treatment of the subject is seriously lacking without addressing the key issues like sustainability in relation to the current most closely related works on this topic. Otherwise, it is not clear what kind of advances this yet another malware detection work makes.

Author Response

All the above has been addressed in the paper and it is highlighted with yellow and we did our best to cover all his questions.
